# Zero-Shot Learning of Causal Models

## Abstract

With the increasing acquisition of datasets over time, we now have access to precise and varied descriptions of the world, encompassing a broad range of phenomena. These datasets can be seen as observations from an unknown causal generative processes, commonly described by Structural Causal Models (SCMs). Recovering SCMs from observations poses formidable challenges, and often require us to learn a specific generative model for each dataset. In this work, we propose to learn a *single* model capable of inferring the SCMs in a zero-shot manner. Rather than learning a specific SCM for each dataset, we enable the Fixed-Point Approach (FiP) (Scetbon et al., 2024) to infer the generative SCMs conditionally on their empirical representations. As a by-product, our approach can perform zero-shot generation of new dataset samples and intervened samples. We demonstrate via experiments that our amortized procedure achieves performances on par with SoTA methods trained specifically for each dataset on both in and out-of-distribution problems. To the best of our knowledge, this is the first time that SCMs are inferred in a zero-shot manner from observations, paving the way for a paradigmatic shift towards the assimilation of causal knowledge across datasets.

## 1. Introduction

Learning the causal generative process from observations is a fundamental problem in several scientific domains (Sachs et al., 2005; Foster et al., 2011; Xie et al., 2012), as it offers a comprehensive understanding of the data generation process, and allows for simulating the effect of controlled experiments/interventions. With a learned model of the generative process, one could even accelerate scientific discoveries by

reliably predicting the effects of unseen interventions, eliminating the need for laboratory experiments (Ke et al., 2023; Zhang et al., 2024). Further, understanding the causal mechanisms behind the data generation process helps in robust representation learning as it provides a principled solution to tackle out-of-distribution (OOD) generalization (Arjovsky et al., 2019; Zhang et al., 2020; Schölkopf et al., 2021).

A popular approach for modeling causal processes is the structural causal model (SCM) framework (Peters et al., 2017) where causal mechanisms are modeled via structured functional relationships, and causal structures are given by directed acyclic graphs (DAGs). Since in several practical applications we only have access to observational data, the task of recovering the generative SCM from observations is an important problem in causality (Pearl, 2009). Solving this inverse problem is challenging as both the graph and the functional relationships modeling the causal mechanisms are unknown a priori. Several works have focused on the graph recovery problem by approximating the discrete search space of DAGs (Chickering, 2002; Peters et al., 2014; Deleu et al., 2022), or using continuous optimization objectives (Zheng et al., 2018; Lachapelle et al., 2019; Lippe et al., 2021). Another line of work has studied the recovery of the functional relationships from data, often under structural assumptions like known causal graphs or topological orders, using maximum likelihood estimation (MLE) independently per node (Blöbaum et al., 2022), autoregressive flows to model SCMs (Khemakhem et al., 2021; Geffner et al., 2022; Javaloy et al., 2023) , or transformers to model SCMs as fixed-point (FiP) problems (Scetbon et al., 2024).

Despite these advances, a major limitation remains: each new dataset of observations requires training a specific model, which prevents sharing of causal knowledge across datasets. This can be alleviated via amortized learning (Gordon et al., 2018; Amos, 2022) as it allows knowledge sharing across datasets through a supervised training objective. Rather than optimizing the parameters of a specific model for each dataset, amortized methods aim at training a *single* model that learns to predict the solution to various instances of the same optimization problem by exploiting their shared structure. Once trained, such methods enables zero-shot inference (without updating parameters) to new problems at test time, a first step towards developing foundation models for causal reasoning. Recent works have proposed tech-

[1]Anonymous Institution, Anonymous City, Anonymous Region, Anonymous Country. Correspondence to: Anonymous Author <anon.email@domain.com>.

Preliminary work. Under review by the International Conference on Machine Learning (ICML). Do not distribute.

niques for amortized causal structure learning (Lorch et al., 2022; Ke et al., 2022), average treatment effect (ATE) estimation (Nilforoshan et al., 2023; Zhang et al., 2023), etc. However, none of these works have yet amortized the learning of the functional relationships to directly infer the SCMs.

**Contributions.** In this work, we introduce a conditional version of FiP (Scetbon et al., 2024), termed Cond-FiP, that zero-shot predicts the functional mechanisms of additive noise SCMs. Our contributions are summarized below.

- We propose a novel framework that enables amortized learning of causal mechanisms across different instances from the functional class of SCMs.
  - To achieve this, we first propose to amortize the learning of an encoder model that aims at inferring the noise from observational data and causal structures, and use its latent representation as embeddings of datasets.
  - Then, to infer causal mechanism, we introduce a novel extension of FiP, termed Cond-FiP, that conditions the fixed-point process on dataset embeddings obtained by our encoder.

- Given a dataset and its causal structure, our approach enables zero-shot generation of new dataset samples, and simulation of intervened ones as well.

- We show empirically that Cond-FiP achieves similar performances as the state-of-the-art (SOTA) approaches trained from scratch for each dataset on both in and out-of-distribution (OOD) problems. Further, Cond-FiP obtains better results than baselines in scare data regimes, due to its amortized training procedure.

## 2. Related Works

**Amortized Causal Learning.** There is an increasing emphasis on developing amortized training methods in causality research. The primary benefit with these methods is their ability to incorporate information from a variety of datasets, and learn general algorithms that can infer infer causal knowledge from observations in a zero-shot manner. Lorch et al. (2022) and Ke et al. (2022) were the earliest works to develop amortized structure learning methods, where they propose transformer-based architectures to integrate samples from multiple datasets. They sample datasets from synthetic data generators during training, which enables them to use supervised objectives for structure learning. This also connects with the recent literature on in-context learning of function classes using transformers (Müller et al., 2021; Akyürek et al., 2022; Garg et al., 2022; Von Oswald et al., 2023). These methods were improved by Wu et al. (2024) for OOD generalization, and applied for the challenging task of recovering gene regulatory networks (Ke et al., 2023).

Further, recent works have developed amortized techniques for ATE estimation (Nilforoshan et al., 2023; Zhang et al., 2023), model selection for causal discovery (Gupta et al., 2023), partial causal discovery (topological order) (Scetbon et al., 2024), etc. In this work, we tackle the novel task of amortized learning to infer the causal mechanisms of SCMs.

**Autoregressive Causal Learning.** While a vast majority of the literature on causal discovery concerns structure learning (Chickering, 2002; Peters et al., 2014; Zheng et al., 2018), recent works on causal autoregressive flows (Khemakhem et al., 2021; Javaloy et al., 2023) focus on SOTA generative modeling techniques for learning the causal generative processes induced by SCMs. Khemakhem et al. (2021) proved a novel connection between SCMs and autoregressive flows, as the mapping from noise variables to observable variables in SCMs is a triangular map given the topological order of the causal graph. While their work restricted the functional relationships to additive and affine flows, this was extended by Javaloy et al. (2023) to more flexible triangular monotonic increasing maps. More recently, Scetbon et al. (2024) proposed to directly model SCMs, viewed as fixed-point problems on the ordered nodes, using transformer-based architectures. While these methods enable efficient learning of SCMs and their generative processes, they all require to train a specific generative model per dataset. In contrast, we present a novel extension of FiP (Scetbon et al., 2024) that allows amortized learning of functional relationships across different instances from the functional class of SCMs.

## 3. Amortized Causal Learning: Setup & Background

**Structural Causal Models.** An SCM defines the causal generative process of a set of $d$ endogenous (causal) random variables $\boldsymbol{V} = \{X_1, \cdots, X_d\}$, where each causal variable $X_i$ is defined as a function of a subset of other causal variables ($\boldsymbol{V} \setminus \{X_i\}$) and an exogenous noise variable $N_i$:

$$X_i = F_i(PA(X_i), N_i) \text{ s.t. } PA(X_i) \subset \boldsymbol{V} , \; X_i \notin PA(X_i) \tag{1}$$

Hence, an SCM describes the data-generation process of $\boldsymbol{X} := [X_1, \cdots, X_d] \sim \mathbb{P}_{\boldsymbol{X}}$ from the noise variables $\boldsymbol{N} := [N_1, \cdots, N_d] \sim \mathbb{P}_{\boldsymbol{N}}$ via the function $\boldsymbol{F} := [F_1, \cdots, F_d]$, and a graph $\boldsymbol{\mathcal{G}} \in \{0,1\}^{d \times d}$ indicating the parents of each variable $X_i$, that is $[\boldsymbol{\mathcal{G}}]_{i,j} := 1$ if $X_j \in PA(X_i)$. $\boldsymbol{\mathcal{G}}$ is assumed to be a directed and acyclic graph (DAG), and SCMs to be Markovian (independent noise variables), denoted as $\mathcal{S}(\mathbb{P}_{\boldsymbol{N}}, \boldsymbol{\mathcal{G}}, \boldsymbol{F})$

In addition, we only consider *additive noise models* (ANM), which are SCMs of the form $X_i = F_i(PA(X_i)) + N_i$.

Figure 1. Sketch of the approach proposed in this work. Given a dataset of observations $D_{\boldsymbol{X}}$ and a causal graph $\mathcal{G}$ obtained from an unkown SCM $\mathcal{S}(\mathbb{P}_{\boldsymbol{N}}, \boldsymbol{\mathcal{G}}, \boldsymbol{F})$, the encoder produces a dataset embedding $\mu(D_{\boldsymbol{X}}, \mathcal{G})$, which serves as a condition to instantiate Cond-FiP. Then for any point $\boldsymbol{z} \in \mathbb{R}^d$, $\mathcal{T}(\boldsymbol{z}, D_{\boldsymbol{X}}, \mathcal{G})$ aims at replicating the functional mechanism $\boldsymbol{F}(\boldsymbol{z})$ of the generative SCM.

**DAG-Attention Mechanism.** In FiP (Scetbon et al., 2024) the authors propose to leverage the transformer architecture to learn SCMs from observations. By reparameterizing an SCM according to a topological ordering induced by its graph, the authors show that any SCM can be reformulated as a fixed-point problem of the form $\boldsymbol{X} = \boldsymbol{H}(\boldsymbol{X}, \boldsymbol{N})$ where $\boldsymbol{H}$ admits a simple triangular structure:

$$[\mathrm{Jac}_{\boldsymbol{x}} \boldsymbol{H}(\boldsymbol{x}, \boldsymbol{n})]_{i,j} = 0, \quad \text{if} \quad j \geq i$$
$$[\mathrm{Jac}_{\boldsymbol{n}} \boldsymbol{H}(\boldsymbol{x}, \boldsymbol{n})]_{i,j} = 0, \quad \text{if} \quad i \neq j,$$

where $\mathrm{Jac}_{\boldsymbol{x}} \boldsymbol{H}$, $\mathrm{Jac}_{\boldsymbol{n}} \boldsymbol{H}$ denote the Jacobian of $\boldsymbol{H}$ w.r.t the first and second variables respectively.

Motivated by this fixed-point reformulation, FiP considers a transformer-based architecture to model the functional relationships of SCMs and propose a new attention mechanism to represent DAGs in a differentiable manner. Recall that the standard attention matrix is defined as:

$$\mathrm{A}_{\boldsymbol{M}}(\boldsymbol{Q}, \boldsymbol{K}) = \frac{\exp((\boldsymbol{Q}\boldsymbol{K}^T - \boldsymbol{M})/\sqrt{d_h})}{\exp((\boldsymbol{Q}\boldsymbol{K}^T - \boldsymbol{M})/\sqrt{d_h}) \, \mathbf{1}_d} \quad (2)$$

where $\boldsymbol{Q}, \boldsymbol{K} \in \mathbb{R}^{d \times d_h}$ denote the keys and queries for a single attention head, and $\boldsymbol{M} \in \{0, +\infty\}^{d \times d}$ is a (potential) mask. When M is chosen to be a triangular mask, the attention mechanism (2) enables to parameterize the effects of previous nodes on the current one However, the normalization inherent to the softmax operator in standard attention mechanisms prevents effective modeling of root nodes, which *should not* be influenced by any other node in the graph. To alleviate this issue, FiP proposes to consider the following formulation instead:

$$\mathrm{DA}_{\boldsymbol{M}}(\boldsymbol{Q}, \boldsymbol{K}) = \frac{\exp((\boldsymbol{Q}\boldsymbol{K}^T - \boldsymbol{M})/\sqrt{d_h})}{\mathcal{V}(\exp((\boldsymbol{Q}\boldsymbol{K}^T - \boldsymbol{M})/\sqrt{d_h}) \, \mathbf{1}_d)} \quad (3)$$

where $\mathcal{V}_i(\boldsymbol{v}) = v_i$ if $v_i \geq 1$, else $\mathcal{V}_i(\boldsymbol{v}) = 1$ for any $\boldsymbol{v} \in \mathbb{R}^d$. While softmax forces the coefficients along each row of the attention matrix to sum to one, the attention mechanism described in (3) allows the rows to sum in $[0, 1]$, thus enabling to model root nodes in attention.

**Amortized Learning of SCMs.** The goal of amortized learning is to learn (meta-learning) to predict the solution of similar instances of the same optimization problem (Gordon et al., 2018). More formally, given some random inputs $\mathbf{I} \sim P_{\mathbf{I}}$ and an objective function $L(\theta, \mathbf{I})$, the goal of amortized learning is to learn a parameterized model $\mathcal{M}_\phi$ s.t.

$$\mathcal{M}_\phi(\mathbf{I}) \simeq \theta^*(\mathbf{I}) := \mathrm{argmin}_{\theta \in \Theta} L(\theta, \mathbf{I})$$

To train such a model, amortized learning requires to have access to various pairs $(\mathbf{I}, \theta^*(\mathbf{I}))$ and optimize the parameters $\phi$ of $\mathcal{M}$ in a supervised manner (Garg et al., 2022).

In our work, we aim to develop amortized learning techniques to predict functional mechanisms from datasets (and causal graphs). More formally, in our setting, $\mathbf{I} := (D_{\mathbf{X}}, \mathcal{G})$ and $\theta^*(\mathbf{I}) := \boldsymbol{F}$ where $\boldsymbol{F}$ is the true SCM generating $D_{\mathbf{X}}$. Hence, our goal is to learn a single model capable of zero-shot inference of generative SCMs at test time.

## 4. Methodology: Conditional FiP

In this section, we introduce the proposed approach, Cond-FiP, composed of two key components: (1) a dataset encoder that generates dataset embeddings based on observations and causal graphs, and (2) a conditional variant of FiP (Scetbon et al., 2024), designed for zero-shot inference of the SCMs of datasets when conditioned on their embeddings produced by the encoder. We first present our dataset encoder, then introduce cond-FiP, and conclude with details on generating new observational and interventional samples.

## 4.1. Dataset Encoder

The objective of this section is to develop a method capable of producing efficient latent representations of datasets. To achieve this, we propose to train an encoder that predicts the noise samples from their associated observations given the causal structures using amortized learning. The latent representations obtained by our encoder will be used later as dataset embeddings to infer SCMs in a zero-shot manner.

**Training Setting.** We consider the amortized setting where at training time, we have access to empirical representations of $K$ SCMs $\left(\mathcal{S}(\mathbb{P}_{\boldsymbol{N}}^{(k)}, \boldsymbol{\mathcal{G}}^{(k)}, \boldsymbol{F}^{(k)})\right)_{k=1}^{K}$ that have been sampled independently according to a distribution of SCMs $\mathcal{S}(\mathbb{P}_{\boldsymbol{N}}^{(k)}, \boldsymbol{\mathcal{G}}^{(k)}, \boldsymbol{F}^{(k)}) \sim \mathbb{P}_{\mathcal{S}}$. These empirical representations, denoted $(D_{\boldsymbol{X}}^{(k)}, \boldsymbol{\mathcal{G}}^{(k)})_{k=1}^{K}$ respectively, contain each $n$ observations $D_{\boldsymbol{X}}^{(k)} := [\boldsymbol{X}_1^{(k)}, \ldots, \boldsymbol{X}_n^{(k)}]^T \in \mathbb{R}^{n \times d}$, and the causal graph $\boldsymbol{\mathcal{G}}^{(k)} \in \{0,1\}^{d \times d}$. At training time, we also require to have access to the associated noise samples $D_{\boldsymbol{N}}^{(k)} := [\boldsymbol{N}_1^{(k)}, \ldots, \boldsymbol{N}_n^{(k)}]^T \in \mathbb{R}^{n \times d}$, which play the role of the target variable in our supervised task. For the sake of clarity, we will omit the dependence on $k$ in our notation and assume access to the full distribution of SCMs $\mathbb{P}_{\mathcal{S}}$. Our goal here is to learn a model that given a dataset of observations $D_{\boldsymbol{X}}$ and the causal graph associated $\boldsymbol{\mathcal{G}}$, recovers the true noise $D_{\boldsymbol{N}}$ from which the observations have been generated. By solving this prediction task, we expect the trained model to provide efficient dataset embeddings as detailed below.

**Encoder Architecture.** Following (Lorch et al., 2021; Scetbon et al., 2024), we propose to encode datasets using a transformer-based architecture that alternatively attends on both the sample and node dimensions of the input. More specifically, after having embedded the dataset $D_{\boldsymbol{X}}$ into a higher dimensional space using a linear operation $L(D_{\boldsymbol{X}}) \in \mathbb{R}^{n \times d \times d_h}$ where $d_h$ is the hidden dimension, the encoder $E$ alternates the application of transformer blocks, consisting of a self-attention block followed by an MLP block (Vaswani et al., 2017), where the attention mechanism is applied either across the samples $n$ or the nodes $d$. When attending over samples, the encoder uses standard self-attention as defined in (2) without masking ($\boldsymbol{M} = \{0\}^{n \times n}$), however, when the model attends over the nodes, we leverage the knowledge of the causal graph to mask the undesirable relationships between the nodes, that is we set $\boldsymbol{M} = +\infty \times (1 - \boldsymbol{\mathcal{G}})$, with the convention that $0 \times (+\infty) = 0$, in the standard attention (2). Finally, the obtained embeddings $E(L(D_{\boldsymbol{X}}), \boldsymbol{\mathcal{G}}) \in \mathbb{R}^{n \times d \times d_h}$ are passed to a prediction network $H : \mathbb{R}^{n \times d \times d_h} \to \mathbb{R}^{n \times d}$ defined as 2-hidden layers MLP which brings back the encoded datasets to their original space.

**Training Procedure.** To infer the noise samples in a zero-shot manner, we propose to minimize the mean squared error (MSE) of predicting the target noises $D_{\boldsymbol{N}}$ from the input $(D_{\boldsymbol{X}}, \boldsymbol{\mathcal{G}})$ over the distribution of SCMs $\mathbb{P}_{\mathcal{S}}$ available during training:

$$\mathbb{E}_{\mathcal{S} \sim \mathbb{P}_{\mathcal{S}}} ||D_{\boldsymbol{N}} - H \circ E(L(D_{\boldsymbol{X}}), \boldsymbol{\mathcal{G}})||_2^2 \ .$$

Further, as we restrict ourselves to the case of ANMs, we can equivalently reformulate our training objective in order to predict the functional relationships rather than the noise samples. Indeed, recall that for an ANM $\mathcal{S}(\mathbb{P}_{\boldsymbol{N}}, \boldsymbol{\mathcal{G}}, \boldsymbol{F})$, we have by definition that $\boldsymbol{F}(\boldsymbol{X}) = \boldsymbol{X} - \boldsymbol{N}$. Therefore, by denoting the new targets as $\boldsymbol{F}(D_{\boldsymbol{X}}) := D_{\boldsymbol{X}} - D_{\boldsymbol{N}}$, we propose instead to train our encoder to predict the evaluations of the functional relationships over the SCM distribution by minimizing:

$$\mathbb{E}_{\mathcal{S} \sim \mathbb{P}_{\mathcal{S}}} ||\boldsymbol{F}(D_{\boldsymbol{X}}) - H \circ E(L(D_{\boldsymbol{X}}), \boldsymbol{\mathcal{G}})||_2^2 \ .$$

Please check Appendix A.1 for more details on the encoder training procedure and justification for why recoering noise is equivalent to learning the inverse SCM.

**Inference.** Given a new dataset $D_{\boldsymbol{X}}$ and its causal graph $\boldsymbol{\mathcal{G}}$, the proposed encoder is able to both provide an embedding $E(L(D_{\boldsymbol{X}}), \boldsymbol{\mathcal{G}}) \in \mathbb{R}^{n \times d \times d_h}$, and evaluate the functional mechanisms $\widehat{F}(D_{\boldsymbol{X}}) := H \circ E(L(D_{\boldsymbol{X}}), \boldsymbol{\mathcal{G}})$. However, this model alone is insufficient for generating new data, whether observational or interventional. This task will be addressed by our conditional decoder, which is detailed in the following section.

**On the Knowledge of Graphs.** Prior works on amortized causal learning (Lorch et al., 2022; Ke et al., 2022) proposed to learn to predict the causal graphs using only observations $D_{\boldsymbol{X}}$ on synthetically generated datasets. These methods justify our setup where the true graphs are provided as part of the input context to the model, as we can use them to infer the causal graphs in a zero-shot manner from observations if the true graphs information was not available.

## 4.2. Cond-FiP: Conditional Fixed-Point Decoder

In this section, we present our proposed approach for zero-shot infernce the functional mechanisms of SCMs via amortized training using synthetically generated datasets. To do so, we propose to extend the formulation of FiP with dataset embeddings $E(L(D_{\boldsymbol{X}}), \boldsymbol{\mathcal{G}})$ obtained by our trained encoder as conditions to infer the correct functional mechanisms of the associated SCMs. See Figure 1 for a sketch of Cond-FiP.

**Training Setting.** Analogous to the encoder training setup, we assume that we have access to a distribution of SCMs $\mathcal{S}(\mathbb{P}_{\boldsymbol{N}}, \boldsymbol{\mathcal{G}}, \boldsymbol{F}) \sim \mathbb{P}_{\mathcal{S}}$ at training time, from which we can extract empirical representations of the form $(D_{\boldsymbol{X}}, \boldsymbol{\mathcal{G}})$ containing the observations and the associated causal graphs respectively. Here, we aim at learning a *single* model $\mathcal{T}$ that can do zero-shot inference of the functional mechanisms of an SCM

given its empirical representation. Formally, we aim at training $\mathcal{T}$ such that for given any dataset $D_{\boldsymbol{X}}$ and its associated causal graph $\mathcal{G}$ obtained from an SCM $\mathcal{S}(\mathbb{P}_{\boldsymbol{N}}, \mathcal{G}, \boldsymbol{F}) \sim \mathbb{P}_{\mathcal{S}}$, the conditional function $\boldsymbol{z} \in \mathbb{R}^d \rightarrow \mathcal{T}(\boldsymbol{z}, D_{\boldsymbol{X}}, \mathcal{G}) \in \mathbb{R}^d$ induced by the model approximates the true functional relationship $\boldsymbol{F} : \boldsymbol{z} \in \mathbb{R}^d \rightarrow \boldsymbol{F}(\boldsymbol{z}) \in \mathbb{R}^d$. We achieve this by enabling FiP to be conditioned on dataset embeddings from our dataset encoder, detailed below.

**Decoder Architecture.** The design of our decoder is based on the FiP architecture for fixed-point SCM learning, with two major differences: (1) we use the dataset embeddings obtained from our encoder as a high dimensional codebook to embed the nodes, and (2) we leverage adaptive layer norm operators (Peebles & Xie, 2023) in the transformer blocks of FiP to enable conditional attention mechanisms.

**Conditional Embedding.** The key change of our decoder compared to the original FiP is in the embedding of the input. FiP proposes to embed a data point $\boldsymbol{z} := [z_1, \ldots, z_d] \in \mathbb{R}^d$ into a high dimensional space using a learnable codebook $\boldsymbol{C} := [C_1, \ldots, C_d]^T \in \mathbb{R}^{d \times d_h}$ and positional embedding $\boldsymbol{P} := [P_1, \ldots, P_d]^T \in \mathbb{R}^{d \times d_h}$, from which they define:

$$\boldsymbol{z}_{\text{emb}} := [z_1 * C_1, \ldots, z_d * C_d]^T + \boldsymbol{P} \in \mathbb{R}^{d \times d_h} .$$

By doing so, (Scetbon et al., 2024) ensures that the embedded samples admit the same causal structure as the original samples. However, this embedding layer is only adapted if the samples considered are all drawn from the same observational distribution, as the representation of the nodes, that is given by the codebook $\boldsymbol{C}$, is fixed. In order to generalize their embedding strategy to the case where multiple SCMs are considered, we consider conditional codebooks and positional embeddings adapted for each dataset. Formally, given a dataset $D_{\boldsymbol{X}}$ and a causal graph $\mathcal{G}$, we propose to define the conditional codebook and positional embedding as

$$\boldsymbol{C}(D_{\boldsymbol{X}}, \mathcal{G}) := \mu(D_{\boldsymbol{X}}, \mathcal{G}) W_{\boldsymbol{C}}$$
$$\boldsymbol{P}(D_{\boldsymbol{X}}, \mathcal{G}) := \mu(D_{\boldsymbol{X}}, \mathcal{G}) W_{\boldsymbol{P}}$$

where $\mu(D_{\boldsymbol{X}}, \mathcal{G}) := \text{MaxPool}(E(L(D_{\boldsymbol{X}}), \mathcal{G})) \in \mathbb{R}^{d \times d_h}$ is obtained by max-pooling w.r.t the sample dimension the dataset embedding $E(L(D_{\boldsymbol{X}}), \mathcal{G}) \in \mathbb{R}^{n \times d \times d_h}$ produced by our trained encoder, and $W_{\boldsymbol{C}}, W_{\boldsymbol{P}} \in \mathbb{R}^{d_h \times d_h}$ are learnable parameters. Then we propose to embed any point $\boldsymbol{z} \in \mathbb{R}^d$ conditionally on $(D_{\boldsymbol{X}}, \mathcal{G})$ by considering:

$$\boldsymbol{z}_{\text{emb}} := [z_1 * C_1(D_{\boldsymbol{X}}, \mathcal{G}), \ldots, z_d * C_d(D_{\boldsymbol{X}}, \mathcal{G})]^T$$
$$+ \boldsymbol{P}(D_{\boldsymbol{X}}, \mathcal{G}) \in \mathbb{R}^{d \times d_h}$$

**Adaptive Transfomer Block.** Once an input $\boldsymbol{z} \in \mathbb{R}^d$ has been embedded into a higher dimensional space $\boldsymbol{z}_{\text{emb}} \in \mathbb{R}^{d \times d_h}$, FiP proposes to model SCMs by simulating the reconstruction of the data from noise. Starting from $\boldsymbol{n}_0 \in$

$\mathbb{R}^{d \times d_h}$ a learnable parameter, they propose to update the current noise $L \geq 1$ times by computing:

$$\boldsymbol{n}_{\ell+1} = h(\text{DA}_{\boldsymbol{M}}(\boldsymbol{n}_\ell, \boldsymbol{z}_{\text{emb}}) \boldsymbol{z}_{\text{emb}} + \boldsymbol{n}_\ell)$$

where $h$ refers to the MLP block, and for clarity, we omit both the layer's dependence on its parameters and the inclusion of layer normalization in the notation. Note that here the authors consider their DAG-Attention (3) mechanism in order to modelize correctly the root nodes of the SCM. To obtain a conditional formulation of their computational scheme, we propose first to replace the starting noise $\boldsymbol{n}_0$ by a conditional one w.r.t. $(D_{\boldsymbol{X}}, \mathcal{G})$ and defined as $\boldsymbol{n}_0 := \mu(D_{\boldsymbol{X}}, \mathcal{G}) W_{\boldsymbol{n}_0} \in \mathbb{R}^{d \times d_h}$, where $W_{\boldsymbol{n}_0} \in \mathbb{R}^{d_h \times d_h}$ is a learnable parameter. Additionally, we propose to add adaptive layer normalization operators (Peebles & Xie, 2023) to both attention and MLP blocks, where each scale or shift is obtained by applying a 1 hidden-layer MLP to the condition, which in our case is $\mu(D_{\boldsymbol{X}}, \mathcal{G})$.

**Projection.** To project back the latent representation of $\boldsymbol{z}$ obtained from previous stages, that is $\boldsymbol{n}_L \in \mathbb{R}^{d \times d_h}$, we propose to simply use a linear operation to get $\widehat{\boldsymbol{z}} = \boldsymbol{n}_L W_{\text{out}} \in \mathbb{R}^d$, where $W_{\text{out}} \in \mathbb{R}^{d_h}$ is learnable. In the following, we denote the overall architecture $\mathcal{T}$, which given an input $\boldsymbol{z} \in \mathbb{R}^d$ and some condition $(D_{\boldsymbol{X}}, \mathcal{G})$ outputs $\widehat{\boldsymbol{z}}$, that is $\mathcal{T}(\boldsymbol{z}, D_{\boldsymbol{X}}, \mathcal{G}) = \widehat{\boldsymbol{z}}$. Note that, for simplicity, we omit the dependence of $\widehat{\boldsymbol{z}}$ on $(D_{\boldsymbol{X}}, \mathcal{G})$ in the notation.

**Training Procedure.** Recall that our goal is zero-shot inference of the functional mechanisms of SCMs given their empirical representations. Therefore, to train our model $\mathcal{T}$, we propose to minimize the reconstruction error of the true functional mechanisms estimated by our model over the distribution of SCMs $\mathbb{P}_{\mathcal{S}}$. More precisely, for any SCM $\mathcal{S}(\mathbb{P}_{\boldsymbol{N}}, \mathcal{G}, \boldsymbol{F}) \sim \mathbb{P}_{\mathcal{S}}$ and its empirical representation $(D_{\boldsymbol{X}}, \mathcal{G})$, we aim at minimizing

$$\mathbb{E}_{\boldsymbol{z} \sim \mathbb{P}_{\boldsymbol{X}}} \|\mathcal{T}(\boldsymbol{z}, D_{\boldsymbol{X}}, \mathcal{G}) - \boldsymbol{F}(\boldsymbol{z})\|_2^2 \qquad (4)$$

where $\boldsymbol{z} \sim \mathbb{P}_{\boldsymbol{X}}$ is chosen independent of the random dataset $D_{\boldsymbol{X}}$. Therefore, when integrating over the distribution of SCMs, we obtain the following optimization problem:

$$\mathbb{E}_{\mathcal{S} \sim \mathbb{P}_{\mathcal{S}}} \mathbb{E}_{\boldsymbol{z} \sim \mathbb{P}_{\boldsymbol{X}}} \|\mathcal{T}(\boldsymbol{z}, D_{\boldsymbol{X}}, \mathcal{G}) - \boldsymbol{F}(\boldsymbol{z})\|_2^2 .$$

To compute (4), we propose to sample $n$ independent samples $\boldsymbol{X}'_1, \ldots, \boldsymbol{X}'_n$ from $\mathbb{P}_{\boldsymbol{X}}$, leading to a new dataset $D_{\boldsymbol{X}'}$ independent of $D_{\boldsymbol{X}}$, from which we obtain the following optimization problem:

$$\mathbb{E}_{\mathcal{S} \sim \mathbb{P}_{\mathcal{S}}} \|\mathcal{T}(D_{\boldsymbol{X}'}, D_{\boldsymbol{X}}, \mathcal{G}) - \boldsymbol{F}(D_{\boldsymbol{X}'})\|_2^2 .$$

Therefore our training objective aims at learning $\mathcal{T}$ such that for any given empirical representation $(D_{\boldsymbol{X}}, \mathcal{G})$ of an unknown SCM $\mathcal{S}(\mathbb{P}_{\boldsymbol{N}}, \mathcal{G}, \boldsymbol{F}) \sim \mathbb{P}_{\mathcal{S}}$, the conditional function induced by our model, that is $\boldsymbol{z} \rightarrow \mathcal{T}(\boldsymbol{z}, D_{\boldsymbol{X}}, \mathcal{G})$, is close to

the true functional mechanism $\boldsymbol{F}$ in the MSE sense. In the following, we explain in more detail how to use cond-FiP for causal generation and inference tasks.

**Remark on ANM assumption.** Though our method relies on the ANM assumption for encoder training (Appendix A.1), we do not need this assumption for decoder training! An interesting future work would be to consider more general dataset encoding to get rid of this assumption.

### 4.3. Inference with Cond-FiP

We provide a summary of inference procedure with Cond-FiP, please check Appendix A.2 for more details.

**Observational Generation.** Cond-FiP is capable of generating new data samples: given a random vector noise $\boldsymbol{n} \sim \mathbb{P}_{\boldsymbol{N}}$, we can estimate the observational sample associated according to an unknown SCM $\mathcal{S}(\mathbb{P}_{\boldsymbol{N}}, \mathcal{G}, \boldsymbol{F}) \sim \mathbb{P}_{\mathcal{S}}$ as long as we have access to its empirical representation $(D_{\boldsymbol{X}}, \mathcal{G})$. Formally, starting from $\boldsymbol{n}_0 = \boldsymbol{n}$, we infer the associated observation by computing for $\ell = 1, \ldots, d$:

$$\boldsymbol{n}_\ell = \mathcal{T}(\boldsymbol{n}_{\ell-1}, D_{\boldsymbol{X}}, \mathcal{G}) + \boldsymbol{n} . \tag{5}$$

After (at most) $d$ iterations, $\boldsymbol{n}_d$ corresponds to the observational sample associated to the original noise $\boldsymbol{n}$ according to our conditional SCM $\mathcal{T}(\cdot, D_{\boldsymbol{X}}, \mathcal{G})$. To sample noise from $\mathbb{P}_{\boldsymbol{N}}$, we leverage cond-FiP that can estimates noise samples under the ANM assumption by computing $\widehat{D_{\boldsymbol{N}}} := D_{\boldsymbol{X}} - \mathcal{T}(D_{\boldsymbol{X}}, E(L(D_{\boldsymbol{X}}), \mathcal{G})$. From these estimated noise samples, we can efficiently estimate the joint distribution of the noise by computing the inverse cdfs of the marginals as proposed in FiP.

**Interventional Generation.** Cond-FiP also enables the estimation of interventions given an empirical representation $(D_{\boldsymbol{X}}, \mathcal{G})$ of an unkown SCM $\mathcal{S}(\mathbb{P}_{\boldsymbol{N}}, \mathcal{G}, \boldsymbol{F}) \sim \mathbb{P}_{\mathcal{S}}$. To achieve this, we start from a noise sample $\boldsymbol{n}$, and we generate the associated intervened sample $\widehat{\boldsymbol{z}}^{\mathrm{do}}$ by directly modifying the conditional SCM provided by Cond-FiP. More specifically, we modify in place the SCM obtained by Cond-FiP, leading to its interventional version $\mathcal{T}^{\mathrm{do}}(\cdot, D_{\boldsymbol{X}}, \mathcal{G})$. Now, generating an intervened sample can be done by applying the loop defined in (5), starting from $\boldsymbol{n}$ and using the intervened SCM $\mathcal{T}^{\mathrm{do}}(\cdot, D_{\boldsymbol{X}}, \mathcal{G})$ rather than the original one.

## 5. Experiments

### 5.1. Experimental Setup

**Data Generation Process.** We use the synthetic data generation procedure proposed by Lorch et al. (2022) to generate SCMs in our empirical study, as it offers a wide variety of SCMs, making it ideal for amortized training. We have the option to sample graphs from various schemes and noise variables from diverse distributions. Further, we can con-

trol the complexity of causal relationships: either we set them to be linear (*LIN*) functions randomly sampled, or use random fourier features (*RFF*) for generating random non-linear causal relationships. We construct two distribution of SCMs, $\mathbb{P}_{\mathrm{IN}}$, and $\mathbb{P}_{\mathrm{OUT}}$, which vary based on the choice for sampling causal graphs, noise variables, and causal relationships. Please refer to Appendix B.1 for more details.

**Training Datasets.** We randomly sample SCMs from the $\mathbb{P}_{\mathrm{IN}}$ distribution ($\simeq 4e6$ SCMs), and we restrict the total nodes to be $d = 20$ nodes. From each of these SCMs, we extract the causal graph $\mathcal{G}$ and generate $n_{\mathrm{train}} = 400$ observations to obtain $D_{\boldsymbol{X}}$. This procedure is used for generating the training datasets for both amortized training of the dataset encoder and Cond-FiP.

**Test Datasets.** We evaluate the model for both in-distribution and out-of-distribution generalization by sampling datasets from $\mathbb{P}_{\mathrm{IN}}$ and $\mathbb{P}_{\mathrm{OUT}}$ respectively. We split our collection of test datasets into the four following categories. Our test datasets are categorized as follows: LIN **IN** and RFF **IN** where the SCM are sampled from $\mathbb{P}_{\mathrm{IN}}$ with linear and non-linear functional relationships respectively. Similarly, we define LIN **OUT** and RFF **OUT** where the SCMs are sampled from $\mathbb{P}_{\mathrm{OUT}}$ instead.

For each category, we vary total nodes $d \in [10, 20, 50, 100]$ and sample for each dimension $d$ either 6 or 9 SCMs, based on the total possible schemes for sampling the causal graphs, from which we generate $n_{\mathrm{test}} = 800$ observational samples. Hence, we have a total of 120 test datasets, allowing for a comprehensive evaluation of methods. An interesting aspect of our test setup is that we evaluate the model's ability to generalize to larger graphs ($d = 50$, $d = 100$) despite the train datasets containing only graphs with $d = 20$ nodes.

Further, we generate test datasets using a different synthetic data simulator, C-Suite (Geffner et al., 2022), which consists of 9 different configurations for sampling SCMs.

**Model and Training Configuration.** For both the dataset encoder and cond-FiP, we set the embedding dimension to $d_h = 256$ and the hidden dimension of MLP blocks to 512. Both of our transformer-based models contains 4 attention layers and each attention consists of 8 attention heads. The models were trained for a total of $10k$ epochs with the Adam optimizer (Paszke et al., 2017), where we used a learning rate of $1e-4$ and a weight decay of $5e-9$. We also use the EMA implementation of (Karras et al., 2023) to train our models. Each epoch contains $\simeq 400$ randomly generated datasets from the distribution $\mathbb{P}_{\mathrm{IN}}$, which are processed with a batch size of 2 on a single L40 GPU with 48GB memory.

### 5.2. Benchmark of Cond-FiP

**Baselines.** We compare Cond-FiP against FiP (Scetbon et al., 2024), DECI (Geffner et al., 2022), and

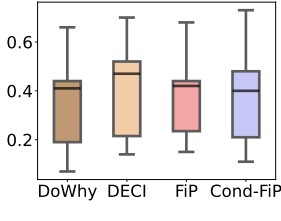

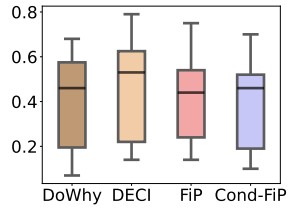

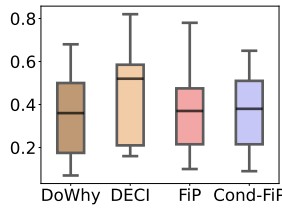

| (a) Noise Prediction | (b) Sample Generation | (c) Interventional Generation |

*Figure 2.* We compare Cond-FiP against the baselines for the different evaluation tasks on the **CSuite benchmark**. The y-axis denotes the RMSE for the respective tasks across the 9 datasets.

| Method | Total Nodes | LIN IN | RFF IN | LIN OUT | RFF OUT |
|--------|-------------|--------|--------|---------|---------|
| DoWhy | 20 | 0.03 | 0.15 | 0.03 | 0.23 |
| DECI | 20 | 0.10 | 0.21 | 0.08 | 0.23 |
| FiP | 20 | 0.04 | 0.12 | 0.05 | 0.15 |
| Cond-FiP | 20 | 0.06 | 0.09 | 0.07 | 0.12 |
| DoWhy | 50 | 0.03 | 0.18 | 0.03 | 0.29 |
| DECI | 50 | 0.09 | 0.24 | 0.07 | 0.29 |
| FiP | 50 | 0.04 | 0.14 | 0.04 | 0.23 |
| Cond-FiP | 50 | 0.06 | 0.10 | 0.07 | 0.14 |
| DoWhy | 100 | 0.03 | 0.20 | 0.03 | 0.31 |
| DECI | 100 | 0.08 | 0.26 | 0.07 | 0.30 |
| FiP | 100 | 0.04 | 0.16 | 0.04 | 0.24 |
| Cond-FiP | 100 | 0.05 | 0.10 | 0.07 | 0.16 |

*Table 1.* **Results for Noise Prediction.** We benchmark Cond-FiP for the task of predicting noise variables from the input observations. Each cell reports the mean RMSE over the multiple test datasets for each scenario. Shaded rows denote the case where the graph size is larger than the train graph sizes ($d = 20$) for Cond-FiP. Please refer to Table 4 for results with standard error.

| Method | Total Nodes | LIN IN | RFF IN | LIN OUT | RFF OUT |
|--------|-------------|--------|--------|---------|---------|
| DoWhy | 20 | 0.06 | 0.27 | 0.05 | 0.39 |
| DECI | 20 | 0.16 | 0.39 | 0.13 | 0.44 |
| FiP | 20 | 0.08 | 0.23 | 0.08 | 0.27 |
| Cond-FiP | 20 | 0.05 | 0.24 | 0.07 | 0.30 |
| DoWhy | 50 | 0.08 | 0.35 | 0.06 | 0.54 |
| DECI | 50 | 0.15 | 0.46 | 0.13 | 0.67 |
| FiP | 50 | 0.09 | 0.26 | 0.08 | 0.48 |
| Cond-FiP | 50 | 0.08 | 0.25 | 0.07 | 0.48 |
| DoWhy | 100 | 0.06 | 0.33 | 0.06 | 0.63 |
| DECI | 100 | 0.14 | 0.50 | 0.14 | 0.71 |
| FiP | 100 | 0.08 | 0.3 | 0.09 | 0.55 |
| Cond-FiP | 100 | 0.07 | 0.29 | 0.09 | 0.57 |

*Table 2.* **Results for Sample Generation.** We benchmark Cond-FiP for the task of generating samples from the input noise variables. Each cell reports the mean RMSE over the multiple test datasets for each scenario. Shaded rows denote the case where the graph size is larger than the train graph sizes ($d = 20$) for Cond-FiP. Please refer to Table 5 for results with standard error.

DoWhy (Blöbaum et al., 2022). We evaluate the zero-shot generalization capabilities of our amortized approach when compared to non-amortized baselines trained from scratch on each test dataset. For a fair comparison with our method, we use $n_{\text{train}} = 400$ samples to train baselines, and evaluate the performance on the remaining 400 test samples. Also, we provide the true graph $\mathcal{G}$ to all the baselines for consistency as Cond-FiP requires true graphs. Finally, we use 400 sample to obtain the dataset embedding, and evaluate Cond-FiP on the remaining ones.

Note that the most important comparison is with the baseline FiP, as Cond-FiP is the amortized version of it. Also, we don't report detailed comparisons with CausalNF (Javaloy et al., 2023), as it didn't provide good results as compared to other baselines (details in Appendix G).

**Evaluation Tasks.** We evaluate the performance of all the methods on the following three tasks.

- **Noise Prediction:** Given the observations $D_X$ and the true graph $\mathcal{G}$, infer the noise variables $\widehat{D_N}$

- **Sample Generation:** Given the noise samples $D_N$ and the true graph $\mathcal{G}$, generate the causal variables $\widehat{D_X}$

- **Interventional Generation:** Generate intervened samples from noise samples $D_N$ and the true graph $\mathcal{G}$.

**Metric.** Let us denote a predicted & true target as $\widehat{Y} \in \mathbb{R}^{n_{\text{test}} \times d}$ and $Y \in \mathbb{R}^{n_{\text{test}} \times d}$. Then RMSE is computed as $\frac{1}{n_{\text{test}}} \sum_{i=1}^{n_{\text{test}}} \sqrt{\frac{1}{d} \|[Y]_i - [\widehat{Y}]_i\|_2^2}$. Note that we scale RMSE by dimension $d$, which allows us to compare results across different graph sizes.

**Results on Noise Predictions.** Table 1 presents the results for the case of inferring noise from observations. Across all the different scenarios (in/out-distribution), Cond-FiP is competitive with the baselines that were trained from scratch at test time. Further, Cond-FiP is able to generalize to larger graphs ($d = 50, d = 100$) despite being trained for only graphs of size $d = 20$. We also obtain similar findings with the CSuite benchmark (Figure 2a), which is a different simulator than what we used for training Cond-FiP.

**Results on Sample Generation.** We now test the generative capabilities of Cond-FiP, where the models are provided with input noise variables. Table 2 presents results for generating observational data, and shows that Cond-FiP is competitive with the baselines across all the scenarios. Similar to the case of noise prediction, Cond-FiP can generalize to larger graphs at test time and is robust to datasets generated from different simulators like CSuite (Figure 2b). We also obtain similar finding a real-world Protein Sachs dataset (Sachs et al., 2005) where we compare the distribution generated by models with the true distribution of data according to the Maximum Mean Discrepency (Gretton et al., 2012). Please refer to Appendix C for more details.

**Results on Interventional Generation.** Cond-FiP can also simulate interventional data while being robust to distribution shifts, graph sizes (Table 3), and different simulators (Figure 2c). This is especially interesting as we never explicitly trained Cond-FiP for interventional tasks. This provide further evidence towards Cond-FiP capturing the true causal/functional mechanisms.

**Results without True Causal Graph.** Our results so far require the knowledge of true graph ($\mathcal{G}$) as part of the input context to Cond-FiP. However, we can extend Cond-FiP to work without access to true causal graph by using prior works (Lorch et al., 2022; Ke et al., 2022) to first infer causal graph in zero-shot manner. Appendix D provides results for experiments where we do not assume the knowledge of true graphs, and we find Cond-FiP can be augmented to zero-shot infer the full SCMs from observations only.

**Scare Data Regime.** An advantage of zero-shot inference methods is their generalization capabilities when the dataset size becomes smaller. As the test dataset size decreases, the performances of baselines can be severely affected since they require training from scratch on these datasets. In contrast, zero-shot inference methods are less impacted as their parameters remains unchanged at inference, and the inductive bias learned during their training phase enables them to generalize even with a smaller input context. Appendix E provides detailed results for this case (illustration for specific setting in Figure 3), we find that *Cond-FiP exhibits superior generalization as compared to the baselines*.

**Ablation Study.** We conduct ablations on the encoder and the decoder to better understand the effect of the training data on the generalization performances. We find that Cond-FiP with encoder trained using only linear/rff data is still competitive to the case of training on combined linear and rff data. However, for decoder, training on combined data is better than training with only linear/rff data. Please refer to Appendix F.1 and F.2 for more details. We also test the sensitivity of Cond-FiP to distribution shifts, comparing its performance across scenarios as the severity of the distribution shift is increased (details in Appendix F.3).

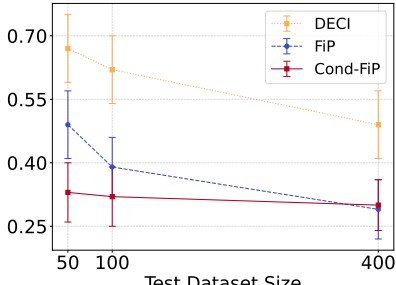

*Figure 3.* We compare the performance (RMSE) of Cond-FiP for **interventional generation** ($d = 100$ and RFF **IN**) as we reduce the test dataset size. Cond-FiP generalizes much better than the baselines in the low-data regime. Detailed results in Appendix E.

| Method | Total Nodes | LIN **IN** | RFF **IN** | LIN **OUT** | RFF **OUT** |
|---|---|---|---|---|---|
| DoWhy | 20 | 0.06 | 0.27 | 0.05 | 0.36 |
| DECI | 20 | 0.16 | 0.38 | 0.15 | 0.42 |
| FiP | 20 | 0.09 | 0.23 | 0.12 | 0.25 |
| Cond-FiP | 20 | 0.09 | 0.24 | 0.14 | 0.31 |
| DoWhy | 50 | 0.08 | 0.29 | 0.06 | 0.53 |
| DECI | 50 | 0.17 | 0.44 | 0.13 | 0.64 |
| FiP | 50 | 0.11 | 0.25 | 0.09 | 0.46 |
| Cond-FiP | 50 | 0.13 | 0.27 | 0.12 | 0.48 |
| DoWhy | 100 | 0.05 | 0.33 | 0.06 | 0.60 |
| DECI | 100 | 0.14 | 0.49 | 0.15 | 0.70 |
| FiP | 100 | 0.08 | 0.29 | 0.10 | 0.54 |
| Cond-FiP | 100 | 0.10 | 0.30 | 0.14 | 0.58 |

*Table 3.* **Results for Interventional Generation.** We benchmark Cond-FiP for generating interventional data from the input noise variables. Each cell reports the mean RMSE over the multiple test datasets for each scenario. Shaded rows denote the case where the graph size is larger than the train graph sizes ($d = 20$) for Cond-FiP. Please refer to Table 6 for results with standard error.

## 6. Conclusion

In this work, we demonstrate that a *single* model can be trained to infer Structural Causal Models (SCMs) in a zero-shot manner through amortized training. Our proposed method, Cond-FiP, not only generalizes effectively to novel SCMs at test time but also remains robust across varying SCM distributions. To the best of our knowledge, this is the first approach to establish the feasibility of learning causal generative models in a foundational manner.

While Cond-FiP generalizes to new instance problems with larger graphs, it is still unable to improve its performances when given larger context sizes at inference (Appendix H). To further enhance the generalization capabilities, we would need to scale both the model and the training data, allowing the model to encounter more complex and diverse contexts. Future work will focus on scaling Cond-FiP to larger problem instances and application to real-world scenarios.

## Impact Statement

This paper presents work whose goal is to advance the field of Machine Learning. There are many potential societal consequences of our work, none which we feel must be specifically highlighted here.

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

# Appendix

## List of Contents

The content in the Appendix has been organized as follows.

# A. Additional Details on Cond-FiP

## A.1. Details on Encoder Training

We provide further details on training the encoder and show how recovering the noise is equivalent to learn the inverse causal generative process. Recall that an SCM is an *implicit* generative model that, given a noise sample $\mathbf{N}$, generates the corresponding observation according to the following fixed-point equation in $\mathbf{X}$

$$\mathbf{X} = F(\mathbf{X}, \mathbf{N})$$

More precisely, to generate the associated observation, one must solve the above fixed-point equation in $\mathbf{X}$ given the noise $\mathbf{N}$. Let us now introduce the following notation that will be instrumental for the subsequent discussion: we denote $F_{\mathbf{N}}(z) : z \rightarrow F(z, \mathbf{N})$.

Due to the specific structure of $F$ (determined by the DAG $\mathcal{G}$ associated with the SCM), the fixed-point equation mentioned above can be efficiently solved by iteratively applying the function $F_{\mathbf{N}}$ to the noise (see Eq. (5) in the manuscript). As a direct consequence, the observation $\mathbf{X}$ can be expressed as a function of the noise:

$$\mathbf{X} = F_{\text{gen}}(\mathbf{N})$$

where $F_{\text{gen}}(\mathbf{N}) := (F_{\mathbf{N}})^{\circ d}(\mathbf{N})$, $d$ is the number of nodes, and $\circ$ denotes the composition operation. In the following we refer to $F_{\text{gen}}$ as the *explicit* generative model induced by the SCM.

Conversely, assuming that the mapping $z \rightarrow F_{\text{gen}}(z)$ is invertible, then one can express the noise as a function of the data:

$$\mathbf{N} = F_{\text{gen}}^{-1}(\mathbf{X})$$

Therefore, learning to recover the noise from observation is equivalent to learn the function $F_{\text{gen}}^{-1}$, which is exactly the inverse of the explicit generative model $F_{\text{gen}}$. It is also worth noting that under the ANM assumption (i.e. $F(\mathbf{X}, \mathbf{N}) = f(\mathbf{X}) + \mathbf{N}$), $F_{\text{gen}}$ is in fact always invertible and its inverse admits a simple expression which is

$$F_{\text{gen}}^{-1}(z) = z - f(z)$$

Therefore, in this specific case, learning the inverse generative model $F_{\text{gen}}^{-1}$ is exactly equivalent to learning the causal mechanism function $f$.

**Additive Noise Model Assumption.** Our method relies on the ANM assumption only for the training the encoder. This is because we require the encoder to predict the noise from data in order to obtain embeddings, and under the ANM assumption, the mapping from data to noise can be easily expressed as $x \rightarrow x - F(x)$ where $F$ is the generative functional mechanism of the generative ANM. However, if we were to consider general SCMs, i.e. of the form $X = F(X, N)$, we would need access to the mapping $x \rightarrow F^{-1}(x, \cdot)(x)$ (assuming this function is invertible), which for general functions is not tractable. An interesting future work would be to consider a more general dataset encoding (using self-supervised techniques), but we believe this is out of the scope of this work.

## A.2. Inference with Cond-FiP

**Cond-FiP Model.**   Once Cond-FiP is trained, we have access to two trained models: (1) an encoder that given a dataset $D_{\mathbf{X}}$ and the causal graph associated $\mathcal{G}$ produced an embedding $\mu(D_{\mathbf{X}}, \mathcal{G})$ as defined l.277., and (2) a decoder that given an embedding $\mu$ and a causal graph $\mathcal{G}$, produces a function $z \in \mathbb{R}^d \to \mathcal{T}(z, \mu, \mathcal{G}) \in \mathbb{R}^d$.

Cond-FiP simply consists of the composition of the two models, that is given a dataset $D_{\mathbf{X}}$ and the causal graph associated $\mathcal{G}$, produce a function $z \in \mathbb{R}^d \to \mathcal{T}(z, \mu(D_{\mathbf{X}}, \mathcal{G}), \mathcal{G}) \in \mathbb{R}^d$, which for simplicity we denote $z \in \mathbb{R}^d \to \mathcal{T}(z, D_{\mathbf{X}}, \mathcal{G}) \in \mathbb{R}^d$. Having clarified this, we can now proceed to more detailed explanations of the inference process using Cond-FiP.

**Sample Generation.**   Given a dataset $D_{\boldsymbol{X}}$ and its causal graph $\mathcal{G}$, we denote $z \to \mathcal{T}(z, D_{\boldsymbol{X}}, \mathcal{G})$ the function infered by Cond-FiP. This function defines the predicted SCM obtained by our model, and we can directly use it to generate new points. More precisely, given a noise sample $\mathbf{n}$, we can generate the associated observational sample by solving the following equation in $\mathbf{x}$:

$$\mathbf{x} = \mathcal{T}(\mathbf{x}, D_{\boldsymbol{X}}, \mathcal{G}) + \mathbf{n}$$

To solve this fixed-point equation, we rely on the fact that $\mathcal{G}$ is a DAG, which enables to solve the fixed-point problem using the following simple iterative procedure. Starting with $\boldsymbol{z}_0 = \mathbf{n}$, we compute for $\ell = 1, \ldots, d$ where $d$ is the number of nodes

$$\boldsymbol{z}_\ell = \mathcal{T}(\boldsymbol{z}_{\ell-1}, D_{\boldsymbol{X}}, \mathcal{G}) + \mathbf{n}$$

After $d$ iterations we obtain the following,

$$\boldsymbol{z}_d = \mathcal{T}(\boldsymbol{z}_d, D_{\boldsymbol{X}}, \mathcal{G}) + \mathbf{n}$$

Therefore, $\boldsymbol{z}_d$ is the solution of the fixed-point problem above, which corresponds to the observational sample associated to $\mathbf{n}$ according to our predicted SCM $z \to \mathcal{T}(z, D_{\boldsymbol{X}}, \mathcal{G})$.

**Interventional Prediction.**   Recall that given a dataset $D_{\boldsymbol{X}}$ and its causal graph $\mathcal{G}$, $z \in \mathbb{R}^d \to \mathcal{T}(z, D_{\boldsymbol{X}}, \mathcal{G}) \in \mathbb{R}^d$ denotes the SCM infered by Cond-FiP. Let us also denote the coordinate-wise formulation of our SCM defined for any $z \in \mathbb{R}^d$ as $\mathcal{T}(z, D_{\boldsymbol{X}}, \mathcal{G}) = [[\mathcal{T}(z, D_{\boldsymbol{X}}, \mathcal{G})]_1, \ldots, [\mathcal{T}(z, D_{\boldsymbol{X}}, \mathcal{G})]_d]$, where for all $i \in \{1, \ldots, d\}$, $z \in \mathbb{R}^d \to [\mathcal{T}(z, D_{\boldsymbol{X}}, \mathcal{G})]_i \in \mathbb{R}$ is a real-valued function.

In order to intervene on this predicted SCM, we simply have to modify in place the predicted function. For example, assume that we want to perform the following intervention $\mathrm{do}(X_i) = a$. Then, to obtain the intervened SCM, we define a new function $z \to \mathcal{T}^{\mathrm{do}(X_i)=a}(z, D_{\boldsymbol{X}}, \mathcal{G})$ defined for any $z \in \mathbb{R}^d$ as: $[\mathcal{T}^{\mathrm{do}(X_i)=a}(z, D_{\boldsymbol{X}}, \mathcal{G})]_j := [\mathcal{T}(z, D_{\boldsymbol{X}}, \mathcal{G})]_j$ if $j \neq i$ and $[\mathcal{T}^{\mathrm{do}(X_i)=a}(z, D_{\boldsymbol{X}}, \mathcal{G})]_i := a$.

Now, using this intervened SCM $z \to \mathcal{T}^{\mathrm{do}(X_i)=a}(z, D_{\boldsymbol{X}}, \mathcal{G})$, we can apply the exact same generation procedure as the one introduced above to generate intervened samples according to our intervened SCM.

# B. Details on Experiments with AVICI Benchmark

## B.1. Experiment Setup

We use the synthetic data generation procedure proposed by Lorch et al. (2022) to generate SCMs in our empirical study. It provides access to a wide variety of SCMs, hence making it an excellent setting for amortized training.

- **Graphs:** We have the option to sample graphs as per the following schemes: Erods-Renyi (Erdos & Renyi, 1959), scale-free models (Barabási & Albert, 1999), Watts-Strogatz (Watts & Strogatz, 1998), and stochastic block models (Holland et al., 1983).

- **Noise Variables:** To sample noise variables, we can choose from either the gaussian or laplace distribution where variances are sampled randomly.

- **Functional Mechanisms:** We can control the complexity of causal relationships: either we set them to be linear (LIN) functions randomly sampled, or use random fourier features (RFF) for generating random non-linear causal relationships.

We construct two distribution of SCMs $\mathbb{P}_{IN}$, and $\mathbb{P}_{OUT}$, which vary based on the choice for sampling causal graphs, noise variables, and causal relationships. The classification aids in understanding the creation of train and test datasets.

- **In-Distribution ($\mathbb{P}_{\mathbf{IN}}$):** We sample causal graphs using the Erods-Renyi and scale-free models schemes. Noise variables are sampled from the gaussian distribution, and we allow for both LIN and RFF causal relationships.

- **Out-of-Distribution ($\mathbb{P}_{\mathbf{OUT}}$):** Causal graphs are drawn from Watts-Strogatz and stochastic block models schemes. Noise variables follow the laplace distribution, and both the LIN and RFF cases are used to sample functions. However, the parameters of these distributions are sampled from a different range as compared to $\mathbb{P}_{IN}$ to create a distribution shift.

We provide further details on the shift in the support of parameters for functional mechanisms below. For complete details please refer to Table 3, Appendix in Lorch et al. (2022).

- **Linear Functional Mechanism.**
  - *In-Distribution ($\mathbb{P}_{IN}$)*
    * Weights: $\sim U_{\pm}(1, 3)$, Bias $\sim U(-3, 3)$.
  - *Out-of-Distribution ($\mathbb{P}_{OUT}$)*
    * Weights: $\sim U_{\pm}(0.5, 2) \cup U_{\pm}(2, 4)$, Bias $\sim U(-3, 3)$.

- **RFF Functional Mechanism.**
  - *In-Distribution ($\mathbb{P}_{IN}$)*
    * Length Scale: $\sim U(7, 10)$, Output Scale: $\sim U(5, 8) \cup U(8, 12)$, Bias $\sim U_{\pm}(-3, 3)$.
  - *Out-of-Distribution ($\mathbb{P}_{OUT}$):*
    * Length Scale: $\sim U(10, 20)$, Output Scale: $\sim U(8, 12) \cup U(18, 22)$, Bias $\sim U_{\pm}(-3, 3)$.

**B.2. Complete Results for Cond-FiP on AVICI Benchmark**

| Method | Total Nodes | LIN **IN** | RFF **IN** | LIN **OUT** | RFF **OUT** |
|---|---|---|---|---|---|
| DoWhy | 10 | 0.03 (0.0) | 0.13 (0.02) | 0.04 (0.01) | 0.11 (0.01) |
| DECI | 10 | 0.09 (0.01) | 0.23 (0.03) | 0.12 (0.01) | 0.23 (0.03) |
| FiP | 10 | 0.04 (0.0) | 0.09 (0.01) | 0.06 (0.01) | 0.08 (0.01) |
| Cond-FiP | 10 | 0.06 (0.01) | 0.10 (0.01) | 0.07 (0.01) | 0.10 (0.01) |
| DoWhy | 20 | 0.03 (0.01) | 0.15 (0.02) | 0.03 (0.0) | 0.23 (0.01) |
| DECI | 20 | 0.10 (0.02) | 0.21 (0.03) | 0.08 (0.02) | 0.23 (0.02) |
| FiP | 20 | 0.04 (0.0) | 0.12 (0.02) | 0.05 (0.0) | 0.15 (0.02) |
| Cond-FiP | 20 | 0.06 (0.01) | 0.09 (0.01) | 0.07 (0.0) | 0.12 (0.0) |
| DoWhy | 50 | 0.03 (0.0) | 0.18 (0.03) | 0.03 (0.0) | 0.29 (0.03) |
| DECI | 50 | 0.09 (0.01) | 0.24 (0.02) | 0.07 (0.01) | 0.29 (0.02) |
| FiP | 50 | 0.04 (0.0) | 0.14 (0.03) | 0.04 (0.0) | 0.23 (0.04) |
| Cond-FiP | 50 | 0.06 (0.01) | 0.10 (0.01) | 0.07 (0.01) | 0.14 (0.01) |
| DoWhy | 100 | 0.03 (0.0) | 0.20 (0.03) | 0.03 (0.0) | 0.31 (0.02) |
| DECI | 100 | 0.08 (0.02) | 0.26 (0.03) | 0.07 (0.01) | 0.30 (0.02) |
| FiP | 100 | 0.04 (0.0) | 0.16 (0.03) | 0.04 (0.0) | 0.24 (0.02) |
| Cond-FiP | 100 | 0.05 (0.0) | 0.10 (0.01) | 0.07 (0.01) | 0.16 (0.01) |

*Table 4.* **Results for Noise Prediction.** We compare Cond-FiP against the baselines for the task of predicting noise variables from the input observations. Each cell reports the mean (standard error) RMSE over the multiple test datasets for each scenario. Shaded rows denote the case where the graph size is larger than the train graph sizes ($d = 20$) for Cond-FiP.

| Method | Total Nodes | LIN **IN** | RFF **IN** | LIN **OUT** | RFF **OUT** |
|---|---|---|---|---|---|
| DoWhy | 10 | 0.05 (0.0) | 0.18 (0.03) | 0.06 (0.01) | 0.12 (0.02) |
| DECI | 10 | 0.15 (0.02) | 0.33 (0.04) | 0.16 (0.02) | 0.27 (0.03) |
| FiP | 10 | 0.07 (0.0) | 0.13 (0.02) | 0.08 (0.01) | 0.11 (0.02) |
| Cond-FiP | 10 | 0.06 (0.01) | 0.14 (0.02) | 0.05 (0.01) | 0.08 (0.01) |
| DoWhy | 20 | 0.06 (0.01) | 0.27 (0.05) | 0.05 (0.0) | 0.39 (0.04) |
| DECI | 20 | 0.16 (0.02) | 0.39 (0.05) | 0.13 (0.02) | 0.44 (0.04) |
| FiP | 20 | 0.08 (0.01) | 0.23 (0.05) | 0.08 (0.01) | 0.27 (0.04) |
| Cond-FiP | 20 | 0.05 (0.01) | 0.24 (0.06) | 0.07 (0.01) | 0.30 (0.03) |
| DoWhy | 50 | 0.08 (0.01) | 0.35 (0.09) | 0.06 (0.01) | 0.54 (0.06) |
| DECI | 50 | 0.15 (0.01) | 0.46 (0.06) | 0.13 (0.02) | 0.67 (0.06) |
| FiP | 50 | 0.09 (0.01) | 0.26 (0.05) | 0.08 (0.01) | 0.48 (0.06) |
| Cond-FiP | 50 | 0.08 (0.01) | 0.25 (0.05) | 0.07 (0.0) | 0.48 (0.07) |
| DoWhy | 100 | 0.06 (0.0) | 0.33 (0.07) | 0.06 (0.01) | 0.63 (0.07) |
| DECI | 100 | 0.14 (0.02) | 0.50 (0.09) | 0.14 (0.02) | 0.71 (0.08) |
| FiP | 100 | 0.08 (0.01) | 0.3 (0.06) | 0.09 (0.01) | 0.55 (0.08) |
| Cond-FiP | 100 | 0.07 (0.01) | 0.29 (0.07) | 0.09 (0.01) | 0.57 (0.07) |

*Table 5.* **Results for Sample Generation.** We compare Cond-FiP against the baselines for the task of generating samples from the input noise variables. Each cell reports the mean (standard error) RMSE over the multiple test datasets for each scenario. Shaded rows denote the case where the graph size is larger than the train graph sizes ($d = 20$) for Cond-FiP.

| Method | Total Nodes | LIN **IN** | RFF **IN** | LIN **OUT** | RFF **OUT** |
|---|---|---|---|---|---|
| DoWhy | 10 | 0.08 (0.03) | 0.19 (0.04) | 0.05 (0.01) | 0.12 (0.02) |
| DECI | 10 | 0.17 (0.02) | 0.34 (0.04) | 0.13 (0.02) | 0.25 (0.03) |
| FiP | 10 | 0.08 (0.01) | 0.15 (0.02) | 0.07 (0.01) | 0.09 (0.01) |
| Cond-FiP | 10 | 0.10 (0.03) | 0.21 (0.03) | 0.07 (0.01) | 0.11 (0.01) |
| DoWhy | 20 | 0.06 (0.01) | 0.27 (0.06) | 0.05 (0.0) | 0.36 (0.03) |
| DECI | 20 | 0.16 (0.02) | 0.38 (0.05) | 0.15 (0.04) | 0.42 (0.03) |
| FiP | 20 | 0.09 (0.01) | 0.23 (0.05) | 0.12 (0.04) | 0.25 (0.03) |
| Cond-FiP | 20 | 0.09 (0.01) | 0.24 (0.05) | 0.14 (0.03) | 0.31 (0.03) |
| DoWhy | 50 | 0.08 (0.01) | 0.29 (0.05) | 0.06 (0.01) | 0.53 (0.06) |
| DECI | 50 | 0.17 (0.02) | 0.44 (0.06) | 0.13 (0.02) | 0.64 (0.06) |
| FiP | 50 | 0.11 (0.02) | 0.25 (0.05) | 0.09 (0.01) | 0.46 (0.06) |
| Cond-FiP | 50 | 0.13 (0.02) | 0.27 (0.04) | 0.12 (0.02) | 0.48 (0.07) |
| DoWhy | 100 | 0.05 (0.0) | 0.33 (0.07) | 0.06 (0.01) | 0.60 (0.07) |
| DECI | 100 | 0.14 (0.02) | 0.49 (0.08) | 0.15 (0.02) | 0.70 (0.08) |
| FiP | 100 | 0.08 (0.01) | 0.29 (0.07) | 0.10 (0.01) | 0.54 (0.08) |
| Cond-FiP | 100 | 0.10 (0.01) | 0.30 (0.06) | 0.14 (0.02) | 0.58 (0.07) |

*Table 6.* **Results for Interventional Generation.** We compare Cond-FiP against the baselines for the task of generating interventional data from the input noise variables. Each cell reports the mean (standard error) RMSE over the multiple test datasets for each scenario. Shaded rows denote the case where the graph size is larger than the train graph sizes ($d = 20$) for Cond-FiP.

# C. Experiments on Real World Benchmark

We use the real world flow cytometry dataset (Sachs et al., 2005) to benchmark Cond-FiP againts the baselines. This dataset contains $n \simeq 800$ observational samples expressed in a $d = 11$ dimensional space, and the reference (true) causal graph. We sample a train dataset $D_X^{\text{train}} \in \mathbb{R}^{n_{\text{train}} \times d}$ and test dataset $D_X^{\text{test}} \in \mathbb{R}^{n_{\text{test}} \times d}$ of size $n_{\text{train}} = n_{\text{test}} = 400$ each, where the train dataset is to used to train the baselines and obtain dataset embedding for Cond-FiP.

Since we don't have access to the true functional relationships, we cannot compute RMSE for noise prediction or sample generation like we did in our experiments with synthetic benchmarks. Instead for each method, we obtain the noise predictions $\widehat{D_N^{\text{train}}}$ on the train split, and use it to fit a gaussian distribution for each component (node). Then we use the learned gaussian distribution to sample new noise variables, $\widehat{D_N^{\text{sample}}}$, which are mapped to the observations as per the causal mechanisms learned by each method, $\widehat{D_X^{\text{sample}}}$. Finally, we compute the maximum mean discrepancy (MMD) distance between $\widehat{D_X^{\text{sample}}}$ and $D_X^{\text{test}}$ as metric to determine whether the method has captured the true causal relationships. For consistency, we also evaluate the reconstruction performances of the models by using directly the inferred noise $\widehat{D_N^{\text{train}}}$ from the models, and the compute MMD between the reconstructed data and the test data.

Table 7 presents our results, where for reference we also report the MMD distance between the true train and test split, which should be very small since both the datasets are sampled from the same distribution. We find that Cond-FiP is competitive with the baselines that were trained from scratch. Except DoWhy, the MMD distance with reconstructed samples from the methods are close to oracle performance.

| Method | MMD($\widehat{D_X^{\text{sample}}}$, $D_X^{\text{test}}$) | MMD($\widehat{D_X^{\text{train}}}$, $D_X^{\text{test}}$) | MMD($D_X^{\text{train}}$, $D_X^{\text{test}}$) |
|---|---|---|---|
| DoWhy | 0.015 | 0.014 | 0.005 |
| DECI | 0.014 | 0.005 | 0.005 |
| FiP | 0.015 | 0.005 | 0.005 |
| Cond-FiP | 0.013 | 0.005 | 0.005 |

*Table 7.* **Results for Sachs dataset.** We compare Cond-FiP against the baselines for the task of generating sample data on the real world benchmark. Each cell reports the MMD, and we also report the reconstruction error for all of the methods.

# D. Results without True Causal Graph

Our results in the main paper (Table 1, Table 2, Table 3) require the knowledge of true graph ($\mathcal{G}$) as part of the input context to Cond-FiP. In this section we conduct where we don't provide the true graph in the input context, rather we infer the graph $\hat{\mathcal{G}}$ using an amortized causal discovery approach (AVICI (Lorch et al., 2022)) from the observational data $D_{\boldsymbol{X}}$. We chose AVICI for this task since it can enable to infer the graph in a zero-shot manner, hence allowing the combined pipeline of AVICI + Cond-FiP to be a zero-shot technique for learning SCMs. More precisely, AVICI zero-shot infers the graph $\mathcal{G}$ from input context $D_{\boldsymbol{X}}$, and we pass ($\hat{\mathcal{G}}, D_{\boldsymbol{X}}$) as the input context for Cond-FiP. Therefore, for any $\boldsymbol{z} \in \mathbb{R}^d$, Cond-FiP ($\mathcal{T}(\boldsymbol{z}, D_{\boldsymbol{X}}, \hat{\mathcal{G}})$) aims to replicate the functional mechanism $\boldsymbol{F}(\boldsymbol{z})$ of the underlying SCM.

The results for benchmarking Cond-FiP with inferred graphs using AVICI for the task of noise prediction, sample generation, and interventional generation are provided in Table 8, Table 9, and Table 10 respectively. For a fair comparison, the baselines FiP, DECI, and DoWhy also use the inferred graph ($\hat{\mathcal{G}}$) by AVICI instead of the true graph ($\mathcal{G}$). We find that Cond-FiP remains competitive to baselines even for the scenario of unknown true causal graph. Hence, our amortized training procedure can be extended easily for zero-shot inference of both graphs and causal mechanisms of the SCM.

| Method | Total Nodes | LIN **IN** | RFF **IN** | LIN **OUT** | RFF **OUT** |
|---|---|---|---|---|---|
| DoWhy | 10 | 0.16 (0.05) | 0.24 (0.04) | 0.12 (0.03) | 0.12 (0.02) |
| DECI | 10 | 0.21 (0.05) | 0.29 (0.04) | 0.16 (0.03) | 0.19 (0.04) |
| FiP | 10 | 0.16 (0.05) | 0.2 (0.04) | 0.13 (0.03) | 0.09 (0.01) |
| Cond-FiP | 10 | 0.15 (0.05) | 0.2 (0.04) | 0.13 (0.03) | 0.11 (0.01) |
| DoWhy | 20 | 0.19 (0.05) | 0.22 (0.03) | 0.2 (0.03) | 0.26 (0.01) |
| DECI | 20 | 0.23 (0.05) | 0.28 (0.03) | 0.24 (0.04) | 0.28 (0.02) |
| FiP | 20 | 0.2 (0.05) | 0.2 (0.03) | 0.21 (0.03) | 0.21 (0.02) |
| Cond-FiP | 20 | 0.18 (0.05) | 0.17 (0.02) | 0.21 (0.03) | 0.16 (0.02) |
| DoWhy | 50 | 0.44 (0.05) | 0.3 (0.03) | 0.51 (0.03) | 0.38 (0.04) |
| DECI | 50 | 0.46 (0.05) | 0.33 (0.04) | 0.52 (0.03) | 0.42 (0.05) |
| FiP | 50 | 0.44 (0.05) | 0.28 (0.04) | 0.51 (0.03) | 0.35 (0.05) |
| Cond-FiP | 50 | 0.43 (0.05) | 0.24 (0.03) | 0.53 (0.03) | 0.29 (0.04) |
| DoWhy | 100 | 0.49 (0.06) | 0.38 (0.03) | 0.64 (0.03) | 0.53 (0.04) |
| DECI | 100 | 0.5 (0.06) | 0.41 (0.03) | 0.64 (0.03) | 0.55 (0.03) |
| FiP | 100 | 0.49 (0.06) | 0.37 (0.03) | 0.64 (0.03) | 0.51 (0.04) |
| Cond-FiP | 100 | 0.48 (0.06) | 0.34 (0.03) | 0.64 (0.03) | 0.49 (0.04) |

*Table 8.* **Results for Noise Prediction without True Graph.** We compare Cond-FiP against the baselines for the task of predicting noise variable from input observations. Unlike experiments in the main paper, we do not use the true graph, and infer the **graph first in a zero-shot manner** using AVICI (Lorch et al., 2022). Each cell reports the mean (standard error) RMSE over the multiple test datasets for each scenario. Shaded rows deonte the case where the graph size is larger than the train graph sizes ($d = 20$) for Cond-FiP.

| Method | Total Nodes | LIN **IN** | RFF **IN** | LIN **OUT** | RFF **OUT** |
|---|---|---|---|---|---|
| DoWhy | 10 | 0.22 (0.07) | 0.29 (0.05) | 0.13 (0.04) | 0.14 (0.02) |
| DECI | 10 | 0.29 (0.06) | 0.39 (0.05) | 0.18 (0.04) | 0.22 (0.05) |
| FiP | 10 | 0.23 (0.06) | 0.26 (0.05) | 0.15 (0.04) | 0.12 (0.02) |
| Cond-FiP | 10 | 0.22 (0.07) | 0.26 (0.05) | 0.13 (0.04) | 0.11 (0.02) |
| DoWhy | 20 | 0.25 (0.05) | 0.38 (0.06) | 0.29 (0.06) | 0.42 (0.03) |
| DECI | 20 | 0.3 (0.06) | 0.52 (0.07) | 0.34 (0.06) | 0.47 (0.04) |
| FiP | 20 | 0.26 (0.05) | 0.37 (0.07) | 0.3 (0.06) | 0.33 (0.04) |
| Cond-FiP | 20 | 0.24 (0.05) | 0.36 (0.06) | 0.29 (0.06) | 0.35 (0.03) |
| DoWhy | 50 | 0.53 (0.07) | 0.46 (0.06) | 0.58 (0.03) | 0.59 (0.07) |
| DECI | 50 | 0.55 (0.07) | 0.54 (0.07) | 0.59 (0.02) | 0.66 (0.06) |
| FiP | 50 | 0.53 (0.07) | 0.44 (0.05) | 0.58 (0.02) | 0.53 (0.07) |
| Cond-FiP | 50 | 0.52 (0.07) | 0.43 (0.05) | 0.58 (0.02) | 0.53 (0.07) |
| DoWhy | 100 | 0.67 (0.07) | 0.52 (0.06) | 0.69 (0.02) | 0.68 (0.04) |
| DECI | 100 | 0.69 (0.08) | 0.57 (0.08) | 0.69 (0.02) | 0.71 (0.04) |
| FiP | 100 | 0.66 (0.07) | 0.5 (0.07) | 0.68 (0.02) | 0.64 (0.05) |
| Cond-FiP | 100 | 0.64 (0.06) | 0.49 (0.06) | 0.68 (0.02) | 0.63 (0.05) |

*Table 9.* **Results for Sample Generation without True Graph.** We compare Cond-FiP against the baselines for the task of generating samples from the input noise variable. Unlike experiments in the main paper, we do not use the true graph, and infer the **graph first in a zero-shot manner** using AVICI (Lorch et al., 2022). Each cell reports the mean (standard error) RMSE over the multiple test datasets for each scenario. Shaded rows deonte the case where the graph size is larger than the train graph sizes ($d = 20$) for Cond-FiP.

| Method | Total Nodes | LIN **IN** | RFF **IN** | LIN **OUT** | RFF **OUT** |
|---|---|---|---|---|---|
| DoWhy | 10 | 0.32 (0.09) | 0.3 (0.05) | 0.13 (0.04) | 0.13 (0.02) |
| DECI | 10 | 0.37 (0.08) | 0.39 (0.05) | 0.17 (0.03) | 0.21 (0.04) |
| FiP | 10 | 0.32 (0.08) | 0.27 (0.05) | 0.14 (0.04) | 0.1 (0.02) |
| Cond-FiP | 10 | 0.31 (0.08) | 0.3 (0.05) | 0.14 (0.04) | 0.13 (0.02) |
| DoWhy | 20 | 0.29 (0.06) | 0.38 (0.07) | 0.37 (0.05) | 0.4 (0.03) |
| DECI | 20 | 0.34 (0.06) | 0.51 (0.07) | 0.41 (0.05) | 0.43 (0.03) |
| FiP | 20 | 0.3 (0.06) | 0.37 (0.07) | 0.38 (0.05) | 0.31 (0.03) |
| Cond-FiP | 20 | 0.29 (0.06) | 0.37 (0.06) | 0.37 (0.05) | 0.33 (0.03) |
| DoWhy | 50 | 0.54 (0.08) | 0.45 (0.06) | 0.62 (0.04) | 0.57 (0.06) |
| DECI | 50 | 0.57 (0.08) | 0.52 (0.07) | 0.63 (0.03) | 0.64 (0.06) |
| FiP | 50 | 0.55 (0.08) | 0.43 (0.05) | 0.62 (0.03) | 0.51 (0.07) |
| Cond-FiP | 50 | 0.54 (0.08) | 0.43 (0.05) | 0.62 (0.03) | 0.51 (0.06) |
| DoWhy | 100 | 0.66 (0.06) | 0.52 (0.07) | 0.71 (0.05) | 0.65 (0.05) |
| DECI | 100 | 0.68 (0.07) | 0.58 (0.09) | 0.71 (0.05) | 0.7 (0.04) |
| FiP | 100 | 0.65 (0.06) | 0.51 (0.07) | 0.71 (0.05) | 0.62 (0.05) |
| Cond-FiP | 100 | 0.64 (0.06) | 0.49 (0.06) | 0.7 (0.04) | 0.62 (0.05) |

*Table 10.* **Results for Interventional Generation without True Graph.** We compare Cond-FiP against the baselines for the task of interventional data from the input noise variable. Unlike experiments in the main paper, we do not use the true graph, and infer the **graph first in a zero-shot manner** using AVICI (Lorch et al., 2022). Each cell reports the mean (standard error) RMSE over the multiple test datasets for each scenario. Shaded rows deonte the case where the graph size is larger than the train graph sizes ($d = 20$) for Cond-FiP.

# E. Evaluating Generalization of Cond-Fip in Scarce Data Regime

## E.1. Experiments with $n_{\text{test}} = 100$

In this section we benchmark Cond-FiP against the baselines for the scenario when test datasets in the input context have smaller sample size ($n_{\text{test}} = 100$) as compared to the train datasets ($n_{\text{test}} = 400$).

We report the results for the task of noise prediction, sample generation, and interventional generation in Table 11, Table 12, and Table 13 respectively. We find that Cond-FiP exhibits superior generalization as compared to baselines. For example, in the case of RFF **IN**, Cond-FiP is even better than FiP for all the tasks! This can be attributed to the advantage of zero-shot inference; as the sample size in test dataset decreases, the generalization of baselines would be affected a lot since they require training from scratch on these datasets. However, zero-shot inference methods would be impacted less as they do not have to trained from scratch, and the inductive bias learned by them can help them generalize even with smaller input context.

| Method | Total Nodes | LIN **IN** | RFF **IN** | LIN **OUT** | RFF **OUT** |
|---|---|---|---|---|---|
| DoWhy | 10 | 0.06 (0.01) | 0.22 (0.03) | 0.09 (0.01) | 0.16 (0.03) |
| DECI | 10 | 0.15 (0.01) | 0.3 (0.02) | 0.22 (0.01) | 0.3 (0.03) |
| FiP | 10 | 0.07 (0.01) | 0.18 (0.01) | 0.12 (0.01) | 0.11 (0.01) |
| Cond-FiP | 10 | 0.07 (0.01) | 0.14 (0.01) | 0.09 (0.01) | 0.14 (0.01) |
| DoWhy | 20 | 0.06 (0.01) | 0.27 (0.05) | 0.07 (0.01) | 0.37 (0.01) |
| DECI | 20 | 0.15 (0.02) | 0.33 (0.02) | 0.17 (0.02) | 0.35 (0.03) |
| FiP | 20 | 0.09 (0.01) | 0.21 (0.03) | 0.1 (0.01) | 0.27 (0.03) |
| Cond-FiP | 20 | 0.08 (0.01) | 0.12 (0.01) | 0.1 (0.01) | 0.15 (0.01) |
| DoWhy | 50 | 0.06 (0.01) | 0.29 (0.04) | 0.05 (0.01) | 0.47 (0.04) |
| DECI | 50 | 0.14 (0.01) | 0.33 (0.02) | 0.14 (0.02) | 0.4 (0.03) |
| FiP | 50 | 0.08 (0.01) | 0.23 (0.03) | 0.08 (0.01) | 0.37 (0.04) |
| Cond-FiP | 50 | 0.08 (0.0) | 0.12 (0.01) | 0.08 (0.01) | 0.15 (0.01) |
| DoWhy | 100 | 0.06 (0.01) | 0.31 (0.04) | 0.06 (0.01) | 0.5 (0.03) |
| DECI | 100 | 0.13 (0.01) | 0.36 (0.03) | 0.12 (0.02) | 0.44 (0.02) |
| FiP | 100 | 0.08 (0.01) | 0.25 (0.04) | 0.1 (0.01) | 0.39 (0.03) |
| Cond-FiP | 100 | 0.07 (0.0) | 0.13 (0.01) | 0.08 (0.01) | 0.17 (0.01) |

*Table 11.* **Results for Noise Prediction with Smaller Sample Size** ($n_{\text{test}} = 100$). We compare Cond-FiP against the baselines for the task of predicting noise variable from input observations. Each test dataset contains 100 samples, as opposed to 400 samples in the main paper. Each cell reports the mean (standard error) RMSE over the multiple test datasets for each scenario. Shaded rows deonte the case where the graph size is larger than the train graph sizes ($d = 20$) for Cond-FiP.

| Method | Total Nodes | LIN **IN** | RFF **IN** | LIN **OUT** | RFF **OUT** |
|---|---|---|---|---|---|
| DoWhy | 10 | 0.1 (0.01) | 0.3 (0.06) | 0.12 (0.02) | 0.19 (0.03) |
| DECI | 10 | 0.23 (0.01) | 0.45 (0.04) | 0.31 (0.02) | 0.38 (0.04) |
| FiP | 10 | 0.13 (0.01) | 0.29 (0.04) | 0.18 (0.02) | 0.15 (0.03) |
| Cond-FiP | 10 | 0.09 (0.01) | 0.2 (0.03) | 0.09 (0.02) | 0.14 (0.02) |
| DoWhy | 20 | 0.11 (0.01) | 0.47 (0.15) | 0.11 (0.02) | 0.5 (0.03) |
| DECI | 20 | 0.26 (0.02) | 0.53 (0.05) | 0.26 (0.03) | 0.57 (0.04) |
| FiP | 20 | 0.17 (0.02) | 0.34 (0.06) | 0.17 (0.02) | 0.39 (0.03) |
| Cond-FiP | 20 | 0.08 (0.0) | 0.31 (0.06) | 0.13 (0.01) | 0.37 (0.02) |
| DoWhy | 50 | 0.11 (0.01) | 0.42 (0.08) | 0.09 (0.01) | 0.66 (0.06) |
| DECI | 50 | 0.23 (0.02) | 0.59 (0.08) | 0.27 (0.04) | 0.73 (0.06) |
| FiP | 50 | 0.13 (0.01) | 0.38 (0.07) | 0.14 (0.01) | 0.58 (0.06) |
| Cond-FiP | 50 | 0.1 (0.01) | 0.32 (0.05) | 0.12 (0.01) | 0.54 (0.05) |
| DoWhy | 100 | 0.11 (0.01) | 0.44 (0.08) | 0.11 (0.01) | 0.74 (0.05) |
| DECI | 100 | 0.25 (0.02) | 0.62 (0.08) | 0.25 (0.01) | 0.78 (0.07) |
| FiP | 100 | 0.15 (0.01) | 0.4 (0.07) | 0.19 (0.02) | 0.67 (0.07) |
| Cond-FiP | 100 | 0.11 (0.01) | 0.35 (0.07) | 0.14 (0.02) | 0.63 (0.07) |

*Table 12.* **Results for Sample Generation with Smaller Sample Size ($n_{\text{test}} = 100$).** We compare Cond-FiP against the baselines for the task of generating samples from the input noise variable. Each test dataset contains 100 samples, as opposed to 400 samples in the main paper. Each cell reports the mean (standard error) RMSE over the multiple test datasets for each scenario. Shaded rows deonte the case where the graph size is larger than the train graph sizes ($d = 20$) for Cond-FiP.

| Method | Total Nodes | LIN **IN** | RFF **IN** | LIN **OUT** | RFF **OUT** |
|---|---|---|---|---|---|
| DoWhy | 10 | 0.09 (0.01) | 0.34 (0.08) | 0.11 (0.01) | 0.2 (0.04) |
| DECI | 10 | 0.24 (0.02) | 0.43 (0.04) | 0.26 (0.03) | 0.35 (0.04) |
| FiP | 10 | 0.13 (0.01) | 0.29 (0.04) | 0.14 (0.02) | 0.14 (0.03) |
| Cond-FiP | 10 | 0.09 (0.02) | 0.21 (0.03) | 0.09 (0.01) | 0.12 (0.02) |
| DoWhy | 20 | 0.1 (0.01) | 0.37 (0.08) | 0.11 (0.02) | 0.49 (0.04) |
| DECI | 20 | 0.25 (0.03) | 0.5 (0.05) | 0.28 (0.03) | 0.54 (0.04) |
| FiP | 20 | 0.16 (0.01) | 0.33 (0.06) | 0.2 (0.03) | 0.38 (0.03) |
| Cond-FiP | 20 | 0.1 (0.01) | 0.27 (0.05) | 0.15 (0.02) | 0.29 (0.03) |
| DoWhy | 50 | 0.12 (0.02) | 0.49 (0.14) | 0.09 (0.01) | 0.64 (0.07) |
| DECI | 50 | 0.26 (0.03) | 0.56 (0.07) | 0.26 (0.03) | 0.72 (0.06) |
| FiP | 50 | 0.16 (0.02) | 0.36 (0.06) | 0.15 (0.01) | 0.57 (0.06) |
| Cond-FiP | 50 | 0.13 (0.02) | 0.29 (0.04) | 0.12 (0.01) | 0.49 (0.07) |
| DoWhy | 100 | 0.11 (0.01) | 0.46 (0.07) | 0.11 (0.01) | 1.16 (0.38) |
| DECI | 100 | 0.24 (0.02) | 0.62 (0.08) | 0.26 (0.01) | 0.78 (0.07) |
| FiP | 100 | 0.16 (0.02) | 0.39 (0.07) | 0.2 (0.02) | 0.66 (0.07) |
| Cond-FiP | 100 | 0.12 (0.02) | 0.32 (0.07) | 0.13 (0.01) | 0.58 (0.07) |

*Table 13.* **Results for Interventional Generation with Smaller Sample Size ($n_{\text{test}} = 100$).** We compare Cond-FiP against the baselines for the task of generating interventional data from the input noise variable. Each test dataset contains 100 samples, as opposed to 400 samples in the main paper. Each cell reports the mean (standard error) RMSE over the multiple test datasets for each scenario. Shaded rows deonte the case where the graph size is larger than the train graph sizes ($d = 20$) for Cond-FiP.

**E.2. Experiments with $n_{\text{test}} = 50$**

We conduct more experiments for the smaller sample size scenarios, where decrease the sample size even further to $n_{\text{test}} = 50$ samples. We report the results for the task of noise prediction, sample generation, and interventional generation in Table 14, Table 15, and Table 16 respectively. We find that baselines perform much worse than Cond-FiP for the all different SCM distributions, highlighting the efficacy of Cond-FiP for inferring functional relationships when the input context has smaller sample size. Note that there were issues with training DoWhy for such a small dataset, hence we do not consider them for this scenario.

| Method | Total Nodes | LIN **IN** | RFF **IN** | LIN **OUT** | RFF **OUT** |
|---|---|---|---|---|---|
| DECI | 10 | 0.19 (0.02) | 0.41 (0.03) | 0.2 (0.02) | 0.42 (0.04) |
| FiP | 10 | 0.13 (0.03) | 0.27 (0.03) | 0.15 (0.02) | 0.21 (0.03) |
| Cond-FiP | 10 | 0.09 (0.01) | 0.17 (0.01) | 0.11 (0.01) | 0.16 (0.01) |
| DECI | 20 | 0.2 (0.01) | 0.42 (0.03) | 0.25 (0.04) | 0.45 (0.05) |
| FiP | 20 | 0.12 (0.01) | 0.33 (0.04) | 0.15 (0.02) | 0.35 (0.04) |
| Cond-FiP | 20 | 0.1 (0.01) | 0.16 (0.01) | 0.11 (0.01) | 0.17 (0.01) |
| DECI | 50 | 0.2 (0.02) | 0.43 (0.02) | 0.2 (0.03) | 0.5 (0.05) |
| FiP | 50 | 0.13 (0.01) | 0.32 (0.03) | 0.13 (0.01) | 0.49 (0.05) |
| Cond-FiP | 50 | 0.1 (0.01) | 0.16 (0.0) | 0.1 (0.01) | 0.17 (0.01) |
| DECI | 100 | 0.19 (0.02) | 0.43 (0.03) | 0.21 (0.01) | 0.53 (0.02) |
| FiP | 100 | 0.11 (0.01) | 0.32 (0.04) | 0.13 (0.01) | 0.48 (0.02) |
| Cond-FiP | 100 | 0.09 (0.01) | 0.16 (0.01) | 0.09 (0.01) | 0.18 (0.01) |

*Table 14.* **Results for Noise Prediction with Smaller Sample Size** ($n_{\text{test}} = 50$). We compare Cond-FiP against the baselines for the task of predicting noise variable from input observations. Each test dataset contains 50 samples, as opposed to 400 samples in the main paper. Each cell reports the mean (standard error) RMSE over the multiple test datasets for each scenario. Shaded rows deonte the case where the graph size is larger than the train graph sizes ($d = 20$) for Cond-FiP.

| Method | Total Nodes | LIN **IN** | RFF **IN** | LIN **OUT** | RFF **OUT** |
|---|---|---|---|---|---|
| DECI | 10 | 0.31 (0.02) | 0.58 (0.05) | 0.27 (0.04) | 0.49 (0.07) |
| FiP | 10 | 0.2 (0.03) | 0.4 (0.05) | 0.21 (0.03) | 0.25 (0.04) |
| Cond-FiP | 10 | 0.12 (0.02) | 0.28 (0.03) | 0.12 (0.01) | 0.18 (0.03) |
| DECI | 20 | 0.34 (0.02) | 0.66 (0.08) | 0.39 (0.07) | 0.68 (0.05) |
| FiP | 20 | 0.2 (0.01) | 0.51 (0.08) | 0.25 (0.04) | 0.51 (0.02) |
| Cond-FiP | 20 | 0.13 (0.01) | 0.4 (0.06) | 0.19 (0.02) | 0.43 (0.02) |
| DECI | 50 | 0.32 (0.02) | 0.66 (0.06) | 0.36 (0.02) | 0.8 (0.06) |
| FiP | 50 | 0.2 (0.01) | 0.48 (0.07) | 0.22 (0.02) | 0.69 (0.06) |
| Cond-FiP | 50 | 0.15 (0.02) | 0.4 (0.05) | 0.16 (0.01) | 0.59 (0.06) |
| DECI | 100 | 0.36 (0.04) | 0.68 (0.08) | 0.39 (0.03) | 0.84 (0.06) |
| FiP | 100 | 0.2 (0.02) | 0.49 (0.09) | 0.28 (0.03) | 0.73 (0.07) |
| Cond-FiP | 100 | 0.16 (0.01) | 0.42 (0.07) | 0.22 (0.01) | 0.68 (0.06) |

*Table 15.* **Results for Sample Generation with Smaller Sample Size ($n_{\text{test}} = 50$).** We compare Cond-FiP against the baselines for the task of generating samples from the input noise variable. Each test dataset contains 50 samples, as opposed to 400 samples in the main paper. Each cell reports the mean (standard error) RMSE over the multiple test datasets for each scenario. Shaded rows deonte the case where the graph size is larger than the train graph sizes ($d = 20$) for Cond-FiP.

| Method | Total Nodes | LIN **IN** | RFF **IN** | LIN **OUT** | RFF **OUT** |
|---|---|---|---|---|---|
| DECI | 10 | 0.3 (0.03) | 0.53 (0.05) | 0.26 (0.04) | 0.42 (0.05) |
| FiP | 10 | 0.21 (0.04) | 0.35 (0.04) | 0.2 (0.03) | 0.22 (0.03) |
| Cond-FiP | 10 | 0.12 (0.01) | 0.19 (0.03) | 0.07 (0.01) | 0.14 (0.02) |
| DECI | 20 | 0.33 (0.02) | 0.6 (0.06) | 0.43 (0.07) | 0.63 (0.04) |
| FiP | 20 | 0.21 (0.02) | 0.46 (0.07) | 0.29 (0.04) | 0.49 (0.02) |
| Cond-FiP | 20 | 0.11 (0.01) | 0.29 (0.06) | 0.15 (0.02) | 0.32 (0.03) |
| DECI | 50 | 0.34 (0.02) | 0.66 (0.07) | 0.34 (0.02) | 0.78 (0.06) |
| FiP | 50 | 0.21 (0.02) | 0.46 (0.07) | 0.23 (0.02) | 0.68 (0.06) |
| Cond-FiP | 50 | 0.13 (0.02) | 0.31 (0.05) | 0.12 (0.02) | 0.51 (0.07) |
| DECI | 100 | 0.37 (0.04) | 0.67 (0.08) | 0.4 (0.04) | 0.84 (0.06) |
| FiP | 100 | 0.21 (0.02) | 0.49 (0.08) | 0.28 (0.03) | 0.73 (0.07) |
| Cond-FiP | 100 | 0.12 (0.01) | 0.33 (0.07) | 0.14 (0.01) | 0.58 (0.07) |

*Table 16.* **Results for Interventional Generation with Smaller Sample Size ($n_{\text{test}} = 50$).** We compare Cond-FiP against the baselines for the task of generating interventional data from the input noise variable. Each test dataset contains 50 samples, as opposed to 400 samples in the main paper. Each cell reports the mean (standard error) RMSE over the multiple test datasets for each scenario. Shaded rows deonte the case where the graph size is larger than the train graph sizes ($d = 20$) for Cond-FiP.

# F. Ablation Study

## F.1. Ablation Study of Encoder

Similar to the ablation study of decoder Appendix F, we conduct an ablation study where we train two variants of the encoder in Cond-FiP described as follows:

- *Cond-FiP (LIN)*: We sample SCMs with linear functional relationships during training of the encoder.

- *Cond-FiP (RFF)*: We sample SCMs with rff functional relationships during training of the encoder.

Note that for the training the subsequent decoder, we sample SCMs with both linear and rff functional relationships as in the main results ( Table 1, Table 2, and Table 3). Note that in the main results, the encoder was trained by sampling SCMs with both linear and rff functional relationships. Hence, this ablation helps us to understand whether the strategy of training encoder on mixed functional relationships can bring more generalization to the amortization process, or if we should have trained encoders specialized for linear and non-linear functional relationships.

We present our results of the ablation study for the task of noise prediction, sample generation, and interventional generation in Table 17, Table 18, Table 19 respectively. Our findings indicate that Cond-FiP is robust to the choice of encoder training strategy! Even though the encoder for Cond-FiP (LIN) was only trained on data from linear SCMs, its generalization performance is similar to Cond-FiP where the encoder was trained on data from both linear and non-linear SCMs.

| Method | Total Nodes | LIN **IN** | RFF **IN** | LIN **OUT** | RFF **OUT** |
|---|---|---|---|---|---|
| Cond-FiP(LIN) | 10 | 0.07 (0.01) | 0.21 (0.02) | 0.08 (0.01) | 0.2 (0.03) |
| Cond-FiP(RFF) | 10 | 0.06 (0.01) | 0.11 (0.01) | 0.07 (0.01) | 0.09 (0.01) |
| Cond-FiP | 10 | 0.06 (0.01) | 0.1 (0.01) | 0.07 (0.01) | 0.1 (0.01) |
| Cond-FiP(LIN) | 20 | 0.07 (0.01) | 0.19 (0.02) | 0.09 (0.01) | 0.21 (0.01) |
| Cond-FiP(RFF) | 20 | 0.06 (0.01) | 0.09 (0.01) | 0.1 (0.02) | 0.11 (0.01) |
| Cond-FiP | 20 | 0.06 (0.01) | 0.09 (0.01) | 0.07 (0.0) | 0.12 (0.0) |
| Cond-FiP(LIN) | 50 | 0.07 (0.01) | 0.21 (0.02) | 0.07 (0.01) | 0.24 (0.01) |
| Cond-FiP(RFF) | 50 | 0.07 (0.01) | 0.09 (0.01) | 0.07 (0.0) | 0.14 (0.01) |
| Cond-FiP | 50 | 0.06 (0.01) | 0.1 (0.01) | 0.07 (0.01) | 0.14 (0.01) |
| Cond-FiP(LIN) | 100 | 0.06 (0.0) | 0.22 (0.02) | 0.07 (0.01) | 0.26 (0.01) |
| Cond-FiP(RFF) | 100 | 0.06 (0.01) | 0.09 (0.01) | 0.07 (0.01) | 0.14 (0.01) |
| Cond-FiP | 100 | 0.05 (0.0) | 0.1 (0.01) | 0.07 (0.01) | 0.16 (0.01) |

*Table 17.* **Encoder Ablation for Noise Prediction.** We compare Cond-FiP against the baselines for the task of predicting noise variable from input observations against two variants. One variant corresponds to the encoder trained on SCMs with only linear functional relationships, Cond-FiP(LIN). Similarly, we have another variant where the decoder was trained on SCMs with only rff functional relationships, Cond-FiP(RFF). Each cell reports the mean (standard error) RMSE over the multiple test datasets for each scenario.

| Method | Total Nodes | LIN **IN** | RFF **IN** | LIN **OUT** | RFF **OUT** |
|---|---|---|---|---|---|
| Cond-FiP(LIN) | 10 | 0.05 (0.01) | 0.14 (0.02) | 0.06 (0.0) | 0.08 (0.01) |
| Cond-FiP(RFF) | 10 | 0.08 (0.01) | 0.18 (0.06) | 0.06 (0.0) | 0.07 (0.01) |
| Cond-FiP | 10 | 0.06 (0.01) | 0.14 (0.02) | 0.05 (0.01) | 0.08 (0.01) |
| Cond-FiP(LIN) | 20 | 0.05 (0.01) | 0.25 (0.06) | 0.07 (0.01) | 0.3 (0.03) |
| Cond-FiP(RFF) | 20 | 0.08 (0.01) | 0.22 (0.05) | 0.11 (0.01) | 0.29 (0.03) |
| Cond-FiP | 20 | 0.05 (0.01) | 0.24 (0.06) | 0.07 (0.01) | 0.3 (0.03) |
| Cond-FiP(LIN) | 50 | 0.08 (0.01) | 0.26 (0.05) | 0.11 (0.04) | 0.52 (0.08) |
| Cond-FiP(RFF) | 50 | 0.11 (0.01) | 0.26 (0.05) | 0.15 (0.02) | 0.48 (0.07) |
| Cond-FiP | 50 | 0.08 (0.01) | 0.25 (0.05) | 0.07 (0.0) | 0.48 (0.07) |
| Cond-FiP(LIN) | 100 | 0.07 (0.01) | 0.27 (0.06) | 0.08 (0.0) | 0.57 (0.07) |
| Cond-FiP(RFF) | 100 | 0.11 (0.01) | 0.29 (0.08) | 0.18 (0.03) | 0.61 (0.08) |
| Cond-FiP | 100 | 0.07 (0.01) | 0.29 (0.07) | 0.09 (0.01) | 0.57 (0.07) |

*Table 18.* **Encoder Ablation for Sample Generation.** We compare Cond-FiP against the baselines for the task of generating samples from input noise variables against two variants. One variant corresponds to the encoder trained on SCMs with only linear functional relationships, Cond-FiP(LIN). Similarly, we have another variant where the decoder was trained on SCMs with only rff functional relationships, Cond-FiP(RFF). Each cell reports the mean (standard error) RMSE over the multiple test datasets for each scenario.

| Method | Total Nodes | LIN **IN** | RFF **IN** | LIN **OUT** | RFF **OUT** |
|---|---|---|---|---|---|
| Cond-FiP(LIN) | 10 | 0.09 (0.02) | 0.2 (0.03) | 0.06 (0.01) | 0.1 (0.01) |
| Cond-FiP(RFF) | 10 | 0.13 (0.04) | 0.23 (0.08) | 0.08 (0.01) | 0.1 (0.01) |
| Cond-FiP | 10 | 0.1 (0.03) | 0.21 (0.03) | 0.07 (0.01) | 0.11 (0.01) |
| Cond-FiP(LIN) | 20 | 0.08 (0.01) | 0.24 (0.05) | 0.12 (0.04) | 0.3 (0.03) |
| Cond-FiP(RFF) | 20 | 0.13 (0.02) | 0.23 (0.05) | 0.13 (0.03) | 0.31 (0.02) |
| Cond-FiP | 20 | 0.09 (0.01) | 0.24 (0.05) | 0.14 (0.03) | 0.31 (0.03) |
| Cond-FiP(LIN) | 50 | 0.12 (0.02) | 0.29 (0.05) | 0.1 (0.01) | 0.51 (0.07) |
| Cond-FiP(RFF) | 50 | 0.14 (0.02) | 0.29 (0.05) | 0.18 (0.03) | 0.47 (0.06) |
| Cond-FiP | 50 | 0.13 (0.02) | 0.27 (0.04) | 0.12 (0.02) | 0.48 (0.07) |
| Cond-FiP(LIN) | 100 | 0.1 (0.01) | 0.3 (0.06) | 0.12 (0.01) | 0.56 (0.07) |
| Cond-FiP(RFF) | 100 | 0.12 (0.01) | 0.31 (0.07) | 0.2 (0.04) | 0.6 (0.09) |
| Cond-FiP | 100 | 0.1 (0.01) | 0.3 (0.06) | 0.14 (0.02) | 0.58 (0.07) |

*Table 19.* **Encoder Ablation for Interventional Generation.** We compare Cond-FiP against the baselines for the task of generating interventional data from input noise variables against two variants. One variant corresponds to the encoder trained on SCMs with only linear functional relationships, Cond-FiP(LIN). Similarly, we have another variant where the decoder was trained on SCMs with only rff functional relationships, Cond-FiP(RFF). Each cell reports the mean (standard error) RMSE over the multiple test datasets for each scenario.

**F.2. Ablation Study of Decoder**

We conduct an ablation study where we train two variants of the decoder Cond-FiP described as follows:

- *Cond-FiP (LIN):* We sample SCMs with linear functional relationships during training.

- *Cond-FiP (RFF):* We sample SCMs with non-linear functional relationships for training.

Note that in the main results (Tabel 2, Table 3) we show the performances of Cond-FiP trained by sampling SCMs with both linear and non-linear functional relationships. Hence, this ablations helps us to understand whether the strategy of training on mixed functional relationships can bring more generalization to the amortization process, or if we should have trained decoders specialized for linear and non-linear functional relationships.

We present the results of our ablation study in Table 20 and Table 21, for the task of sample generation and interventional generation respectively. Our findings indicate that Cond-FiP decoder trained for both linear and non-linear functional relationships is able to specialize for both the scenarios. While Cond-FiP (LIN) is only able to perform well for linear benchmarks, and similarly Cond-FiP (RFF) can only achieve decent predictions for non-linear benchmarks, Cond-FiP is achieve the best performances on both the linear and non-linear benchmarks.

| Method | Total Nodes | LIN **IN** | RFF **IN** | LIN **OUT** | RFF **OUT** |
|---|---|---|---|---|---|
| Cond-FiP(LIN) | 10 | 0.07 (0.02) | 0.4 (0.06) | 0.07 (0.01) | 0.25 (0.06) |
| Cond-FiP(RFF) | 10 | 0.1 (0.02) | 0.15 (0.02) | 0.08 (0.01) | 0.09 (0.01) |
| Cond-FiP | 10 | 0.06 (0.01) | 0.14 (0.02) | 0.05 (0.01) | 0.08 (0.01) |
| Cond-FiP(LIN) | 20 | 0.07 (0.01) | 0.44 (0.07) | 0.10 (0.01) | 0.58 (0.02) |
| Cond-FiP(RFF) | 20 | 0.11 (0.01) | 0.26 (0.06) | 0.14 (0.01) | 0.31 (0.03) |
| Cond-FiP | 20 | 0.05 (0.01) | 0.24 (0.06) | 0.07 (0.01) | 0.3 (0.03) |
| Cond-FiP(LIN) | 50 | 0.10 (0.01) | 0.5 (0.07) | 0.14 (0.02) | 0.69 (0.04) |
| Cond-FiP(RFF) | 50 | 0.15 (0.02) | 0.27 (0.05) | 0.19 (0.02) | 0.5 (0.07) |
| Cond-FiP | 50 | 0.08 (0.01) | 0.25 (0.05) | 0.07 (0.0) | 0.48 (0.07) |
| Cond-FiP(LIN) | 100 | 0.1 (0.01) | 0.51 (0.07) | 0.15 (0.02) | 0.72 (0.04) |
| Cond-FiP(RFF) | 100 | 0.16 (0.03) | 0.29 (0.07) | 0.27 (0.04) | 0.59 (0.06) |
| Cond-FiP | 100 | 0.07 (0.01) | 0.29 (0.07) | 0.09 (0.01) | 0.57 (0.07) |

*Table 20.* **Decoder Ablation for Sample Generation.** We compare Cond-FiP for the task of generating samples from input noise variables against two variants. One variant corresponds to a decoder trained on SCMs with only linear functional relationships, Cond-FiP(LIN). Similarly, we have another variant where the decoder was trained on SCMs with only rff functional relationships, Cond-FiP(RFF). Each cell reports the mean (standard error) RMSE over the multiple test datasets for each scenario.

| Method | Total Nodes | LIN **IN** | RFF **IN** | LIN **OUT** | RFF **OUT** |
|---|---|---|---|---|---|
| Cond-FiP(**LIN**) | 10 | 0.09 (0.02) | 0.40 (0.07) | 0.06 (0.01) | 0.22 (0.04) |
| Cond-FiP(**RFF**) | 10 | 0.16 (0.05) | 0.22 (0.03) | 0.08 (0.01) | 0.11 (0.01) |
| Cond-FiP | 10 | 0.10 (0.03) | 0.21 (0.03) | 0.07 (0.01) | 0.11 (0.01) |
| Cond-FiP(**LIN**) | 20 | 0.10 (0.01) | 0.45 (0.07) | 0.16 (0.03) | 0.57 (0.02) |
| Cond-FiP(**RFF**) | 20 | 0.14 (0.02) | 0.26 (0.05) | 0.21 (0.03) | 0.32 (0.02) |
| Cond-FiP | 20 | 0.09 (0.01) | 0.24 (0.05) | 0.14 (0.03) | 0.31 (0.03) |
| Cond-FiP(**LIN**) | 50 | 0.14 (0.02) | 0.49 (0.07) | 0.14 (0.02) | 0.68 (0.04) |
| Cond-FiP(**RFF**) | 50 | 0.19 (0.03) | 0.28 (0.05) | 0.21 (0.03) | 0.49 (0.06) |
| Cond-FiP | 50 | 0.13 (0.02) | 0.27 (0.04) | 0.12 (0.02) | 0.48 (0.07) |
| Cond-FiP(**LIN**) | 100 | 0.12 (0.02) | 0.52 (0.07) | 0.18 (0.03) | 0.71 (0.04) |
| Cond-FiP(**RFF**) | 100 | 0.18 (0.03) | 0.32 (0.07) | 0.24 (0.04) | 0.59 (0.07) |
| Cond-FiP | 100 | 0.10 (0.01) | 0.30 (0.06) | 0.14 (0.02) | 0.58 (0.07) |

*Table 21.* **Decoder Ablation for Interventional Generation.** We compare Cond-FiP against two variants for the task of interventional data from input noise variables. One variant corresponds to a decoder trained on SCMs with only linear functional relationships, Cond-FiP(LIN). Similarly, we have another variant where the decoder was trained on SCMs with only rff functional relationships, Cond-FiP(RFF). Each cell reports the mean (standard error) RMSE over the multiple test datasets for each scenario.

### F.3. Sensitivity to Distribution Shifts

In our main results (Table 1, 2, 3) we experimented with datasets sampled from SCM following a different distribution (LIN **OUT**, RFF **OUT**) than the datasets used for training Cond-FiP (LIN **IN**, RFF **IN**). As expected, the performance of all methods drop in the presence of distribution shift. We now analyze how sensitive is Cond-FiP to distribution shifts by comparing its performance across scenarios as the severity of the distribution shift is increased.

To illustrate how we control the magnitude of distribution shift, we discuss the difference in the distribution of causal mechanisms across $\mathbb{P}_{\text{IN}}$ and $\mathbb{P}_{\text{OUT}}$. The distribution shift arises because the support of the parameters of causal mechanisms changes from $\mathbb{P}_{\text{IN}}$ to $\mathbb{P}_{\text{OUT}}$. For example, for linear causal mechanism case, the weights in $\mathbb{P}_{\text{IN}}$ are sampled uniformly from $(-3, -1) \cup (1, 3)$; while in $\mathbb{P}_{\text{OUT}}$ they are sampled from uniformly from $(0.5, 4)$. We now change the support set of the parameters in $\mathbb{P}_{\text{OUT}}$ to $(0.5\alpha, 4\alpha)$, so that by increasing $\alpha$ we make the distribution shift more severe. We follow this procedure for the support set of all the parameters associated with functional mechanisms and generate distributions ($\mathbb{P}_{\text{OUT}}(\alpha)$) with varying shift w.r.t $\mathbb{P}_{\text{IN}}$ by changing $\alpha$. Note that $\alpha = 1$ corresponds to the same $\mathbb{P}_{\text{OUT}}$ as the one used for sampling datasets in our main results.

We conduct two experiments for evaluating the robustness of Cond-FiP to distribution shifts, described ahead.

- **Controlling Shift in Causal Mechanisms.** We start with the parameter configuration of $\mathbb{P}_{\text{OUT}}$ from the setup in main results; and then control the magnitude of shift by changing the support set of parameters of causal mechanisms.

- **Controlling Shift in Noise Variables.** We start with the parameter configuration of $\mathbb{P}_{\text{OUT}}$ from the setup in main results; and then control the magnitude of shift by changing the support set of parameters of noise distribution.

Tables 22, 23, and 24 provide results for the case of controlling shift via causal mechanisms, for the task of noise prediction, sample generation, and interventional generation respectively. We find that the performance of Cond-FiP does not change much as we increase $\alpha$, indicating that Cond-FiP is robust to the varying levels of distribution shits in causal mechanisms.

However, for the case of controlling shift via noise variables (Table 25, 26, and 27) we find that Cond-FiP is quite sensitive to the varying levels of distribution shift in noise variables. The performance of Cond-FiP degrades with increasing magnitude of the shift ($\alpha$) for all the tasks.

| Total Nodes | Shift Level ($\alpha$) | LIN **OUT** | RFF **OUT** |
|:---:|:---:|:---:|:---:|
| 10 | 1 | 0.07 (0.01) | 0.10 (0.01) |
| 10 | 2 | 0.06 (0.01) | 0.10 (0.01) |
| 10 | 5 | 0.05 (0.01) | 0.10 (0.01) |
| 10 | 10 | 0.05 (0.01) | 0.10 (0.01) |
| 20 | 1 | 0.07 (0.0) | 0.12 (0.0) |
| 20 | 2 | 0.06 (0.0) | 0.13 (0.01) |
| 20 | 5 | 0.05 (0.0) | 0.11 (0.01) |
| 20 | 10 | 0.05 (0.0) | 0.10 (0.01) |
| 50 | 1 | 0.07 (0.01) | 0.14 (0.01) |
| 50 | 2 | 0.05 (0.01) | 0.17 (0.01) |
| 50 | 5 | 0.05 (0.01) | 0.14 (0.01) |
| 50 | 10 | 0.04 (0.0) | 0.14 (0.01) |
| 100 | 1 | 0.07 (0.01) | 0.16 (0.01) |
| 100 | 2 | 0.05 (0.01) | 0.18 (0.0) |
| 100 | 5 | 0.05 (0.0) | 0.17 (0.01) |
| 100 | 10 | 0.05 (0.0) | 0.16 (0.01) |

*Table 22.* **Results for Noise Prediction under Distribution Shifts in Causal Mechanisms.** We evaluate the robustness of Cond-FiP to distribution shifts in the parametrization of causal mechanisms. We vary the distribution shift controlled by $\alpha$, where $\alpha = 1$ corresponds to the case in main results Table 1. Each cell reports the mean (standard error) RMSE over the multiple test datasets for each scenario. We find that Cond-FiP is robust to varying levels of distribution shift in causal mechanisms.

| Total Nodes | Shift Level ($\alpha$) | LIN **OUT** | RFF **OUT** |
|:---:|:---:|:---:|:---:|
| 10 | 1 | 0.05 (0.01) | 0.08 (0.01) |
| 10 | 2 | 0.05 (0.0) | 0.07 (0.01) |
| 10 | 5 | 0.05 (0.0) | 0.07 (0.01) |
| 10 | 10 | 0.06 (0.0) | 0.06 (0.01) |
| 20 | 1 | 0.07 (0.01) | 0.30 (0.03) |
| 20 | 2 | 0.06 (0.01) | 0.34 (0.05) |
| 20 | 5 | 0.06 (0.01) | 0.35 (0.05) |
| 20 | 10 | 0.06 (0.01) | 0.29 (0.07) |
| 50 | 1 | 0.07 (0.0) | 0.48 (0.07) |
| 50 | 2 | 0.07 (0.0) | 0.47 (0.07) |
| 50 | 5 | 0.07 (0.01) | 0.38 (0.06) |
| 50 | 10 | 0.07 (0.01) | 0.32 (0.06) |
| 100 | 1 | 0.09 (0.01) | 0.57 (0.07) |
| 100 | 2 | 0.09 (0.01) | 0.60 (0.05) |
| 100 | 5 | 0.09 (0.01) | 0.58 (0.05) |
| 100 | 10 | 0.12 (0.02) | 0.56 (0.06) |

*Table 23.* **Results for Sample Generation under Distribution Shifts in Causal Mechanisms.** We evaluate the robustness of Cond-FiP to distribution shifts in the parametrization of causal mechanisms. We vary the distribution shift controlled by $\alpha$, where $\alpha = 1$ corresponds to the case in main results Table 2. Each cell reports the mean (standard error) RMSE over the multiple test datasets for each scenario. We find that Cond-FiP is robust to varying levels of distribution shift in causal mechanisms.

| Total Nodes | Shift Level ($\alpha$) | LIN **OUT** | RFF **OUT** |
|:---:|:---:|:---:|:---:|
| 10 | 1 | 0.07 (0.01) | 0.11 (0.01) |
| 10 | 2 | 0.07 (0.01) | 0.11 (0.01) |
| 10 | 5 | 0.07 (0.01) | 0.10 (0.01) |
| 10 | 10 | 0.06 (0.01) | 0.10 (0.01) |
| 20 | 1 | 0.14 (0.03) | 0.31 (0.03) |
| 20 | 2 | 0.10 (0.02) | 0.33 (0.04) |
| 20 | 5 | 0.17 (0.1) | 0.34 (0.04) |
| 20 | 10 | 0.10 (0.03) | 0.28 (0.05) |
| 50 | 1 | 0.12 (0.02) | 0.48 (0.07) |
| 50 | 2 | 0.12 (0.03) | 0.47 (0.07) |
| 50 | 5 | 0.11 (0.01) | 0.39 (0.06) |
| 50 | 10 | 0.11 (0.02) | 0.32 (0.06) |
| 100 | 1 | 0.14 (0.02) | 0.58 (0.07) |
| 100 | 2 | 0.13 (0.02) | 0.60 (0.06) |
| 100 | 5 | 0.14 (0.03) | 0.58 (0.05) |
| 100 | 10 | 0.18 (0.04) | 0.55 (0.06) |

*Table 24.* **Results for Interventional Generation under Distribution Shifts in Causal Mechanisms.** We evaluate the robustness of Cond-FiP to distribution shifts in the parametrization of causal mechanisms. We vary the distribution shift controlled by $\alpha$, where $\alpha = 1$ corresponds to the case in main results Table 3. Each cell reports the mean (standard error) RMSE over the multiple test datasets for each scenario. We find that Cond-FiP is robust to varying levels of distribution shift in causal mechanisms.

| Total Nodes | Shift Level ($\alpha$) | LIN **OUT** | RFF **OUT** |
|:-----------:|:----------------------:|:-----------:|:-----------:|
| 10 | 1 | 0.07 (0.01) | 0.10 (0.01) |
| 10 | 2 | 0.07 (0.01) | 0.11 (0.01) |
| 10 | 5 | 0.07 (0.01) | 0.18 (0.02) |
| 10 | 10 | 0.08 (0.01) | 0.26 (0.04) |
| 20 | 1 | 0.07 (0.0) | 0.12 (0.0) |
| 20 | 2 | 0.07 (0.0) | 0.16 (0.01) |
| 20 | 5 | 0.07 (0.0) | 0.30 (0.01) |
| 20 | 10 | 0.07 (0.0) | 0.41 (0.02) |
| 50 | 1 | 0.07 (0.01) | 0.14 (0.01) |
| 50 | 2 | 0.07 (0.01) | 0.19 (0.01) |
| 50 | 5 | 0.07 (0.01) | 0.33 (0.02) |
| 50 | 10 | 0.07 (0.01) | 0.44 (0.02) |
| 100 | 1 | 0.07 (0.01) | 0.16 (0.01) |
| 100 | 2 | 0.07 (0.01) | 0.22 (0.0) |
| 100 | 5 | 0.07 (0.01) | 0.35 (0.01) |
| 100 | 10 | 0.07 (0.01) | 0.44 (0.01) |

*Table 25.* **Results for Noise Prediction under Distribution Shifts in Noise Variables.** We evaluate the robustness of Cond-FiP to distribution shifts in the parametrization of noise distribution. We vary the distribution shift controlled by $\alpha$, where $\alpha = 1$ corresponds to the case in main results Table 1. Each cell reports the mean (standard error) RMSE over the multiple test datasets for each scenario. We find that Cond-FiP is sensitive to varying levels of distribution shift in noise variables, its performance decreases with increasing magnitude of the shift.

| Total Nodes | Shift Level ($\alpha$) | LIN **OUT** | RFF **OUT** |
|:-----------:|:----------------------:|:-----------:|:-----------:|
| 10 | 1 | 0.05 (0.01) | 0.08 (0.01) |
| 10 | 2 | 0.05 (0.0) | 0.13 (0.03) |
| 10 | 5 | 0.05 (0.01) | 0.28 (0.06) |
| 10 | 10 | 0.05 (0.01) | 0.36 (0.08) |
| 20 | 1 | 0.07 (0.01) | 0.30 (0.03) |
| 20 | 2 | 0.07 (0.01) | 0.45 (0.04) |
| 20 | 5 | 0.07 (0.01) | 0.59 (0.03) |
| 20 | 10 | 0.07 (0.01) | 0.58 (0.02) |
| 50 | 1 | 0.07 (0.0) | 0.48 (0.07) |
| 50 | 2 | 0.07 (0.0) | 0.59 (0.06) |
| 50 | 5 | 0.07 (0.0) | 0.64 (0.03) |
| 50 | 10 | 0.07 (0.0) | 0.58 (0.02) |
| 100 | 1 | 0.09 (0.01) | 0.57 (0.07) |
| 100 | 2 | 0.09 (0.01) | 0.63 (0.05) |
| 100 | 5 | 0.09 (0.01) | 0.65 (0.03) |
| 100 | 10 | 0.09 (0.01) | 0.59 (0.02) |

*Table 26.* **Results for Sample Generation under Distribution Shifts in Noise Variables.** We evaluate the robustness of Cond-FiP to distribution shifts in the parametrization of noise distribution. We vary the distribution shift controlled by $\alpha$, where $\alpha = 1$ corresponds to the case in main results Table 2. Each cell reports the mean (standard error) RMSE over the multiple test datasets for each scenario. We find that Cond-FiP is sensitive to varying levels of distribution shift in noise variables, its performance decreases with increasing magnitude of the shift.

| Total Nodes | Shift Level ($\alpha$) | LIN **OUT** | RFF **OUT** |
|:---:|:---:|:---:|:---:|
| 10 | 1 | 0.07 (0.01) | 0.11 (0.01) |
| 10 | 2 | 0.07 (0.01) | 0.14 (0.02) |
| 10 | 5 | 0.07 (0.01) | 0.25 (0.05) |
| 10 | 10 | 0.07 (0.01) | 0.32 (0.06) |
| 20 | 1 | 0.14 (0.03) | 0.31 (0.03) |
| 20 | 2 | 0.14 (0.03) | 0.42 (0.03) |
| 20 | 5 | 0.14 (0.03) | 0.57 (0.03) |
| 20 | 10 | 0.14 (0.03) | 0.56 (0.02) |
| 50 | 1 | 0.12 (0.02) | 0.48 (0.07) |
| 50 | 2 | 0.12 (0.01) | 0.58 (0.06) |
| 50 | 5 | 0.12 (0.01) | 0.65 (0.04) |
| 50 | 10 | 0.12 (0.01) | 0.59 (0.02) |
| 100 | 1 | 0.14 (0.02) | 0.58 (0.07) |
| 100 | 2 | 0.14 (0.02) | 0.65 (0.06) |
| 100 | 5 | 0.14 (0.02) | 0.67 (0.04) |
| 100 | 10 | 0.14 (0.02) | 0.60 (0.03) |

*Table 27.* **Results for Interventional Generation under Distribution Shifts in Noise Variables.** We evaluate the robustness of Cond-FiP to distribution shifts in the parametrization of noise distribution. We vary the distribution shift controlled by $\alpha$, where $\alpha = 1$ corresponds to the case in main results Table 3. Each cell reports the mean (standard error) RMSE over the multiple test datasets for each scenario. We find that Cond-FiP is sensitive to varying levels of distribution shift in noise variables, its performance decreases with increasing magnitude of the shift.

# G. Comparing Cond-FiP with CausalNF

We also compare Cond-FiP with CausalNF (Javaloy et al., 2023) for the task of noise prediction (Table 28) and sample generation (Table 29). The test datasets consist of $n_{\text{test}} = 400$ samples, exact same setup as in our main results (Table 1, Table 2, and Table 3). To ensure a fair comparison, we provided CausalNF with the true causal graph.

Our analysis reveals that CausalNF underperforms compared to Cond-FiP in both tasks, and it is also a weaker baseline relative to FiP. Note also the authors did not experiment with large graphs for CausalNF; the largest graph they used contained approximately 10 nodes. Also, they trained CausalNF on much larger datasets with a sample size of 20k, while our setup has datasets with 400 samples only.

| Method | Total Nodes | LIN **IN** | RFF **IN** | LIN **OUT** | RFF **OUT** |
|---|---|---|---|---|---|
| CausalNF | 10 | 0.16 (0.02) | 0.41 (0.09) | 0.38 (0.04) | 0.35 (0.02) |
| Cond-FiP | 10 | 0.06 (0.01) | 0.10 (0.01) | 0.07 (0.01) | 0.10 (0.01) |
| CausalNF | 20 | 0.18 (0.03) | 0.45 (0.12) | 0.29 (0.05) | 0.36 (0.03) |
| Cond-FiP | 20 | 0.06 (0.01) | 0.09 (0.01) | 0.07 (0.00) | 0.12 (0.00) |
| CausalNF | 50 | 0.25 (0.03) | 0.56 (0.09) | 0.45 (0.06) | 0.38 (0.04) |
| Cond-FiP | 50 | 0.06 (0.01) | 0.10 (0.01) | 0.07 (0.01) | 0.14 (0.01) |
| CausalNF | 100 | 0.24 (0.02) | 0.80 (0.1) | 0.37 (0.06) | 0.49 (0.05) |
| Cond-FiP | 100 | 0.05 (0.0) | 0.10 (0.01) | 0.07 (0.01) | 0.16 (0.01) |

*Table 28.* **Results for Noise Prediction with CausalNF.** We compare Cond-FiP against CausalNF for the task of predicting noise variables from input observations. We find that CausalNF underperforms compared to Cond-FiP by a significant margin.

| Method | Total Nodes | LIN **IN** | RFF **IN** | LIN **OUT** | RFF **OUT** |
|---|---|---|---|---|---|
| CausalNF | 10 | 0.27 (0.07) | 0.29 (0.04) | 0.20 (0.03) | 0.20 (0.03) |
| Cond-FiP | 10 | 0.06 (0.01) | 0.14 (0.02) | 0.05 (0.01) | 0.08 (0.01) |
| CausalNF | 20 | 0.23 (0.02) | 0.36 (0.05) | 0.22 (0.02) | 0.45 (0.02) |
| Cond-FiP | 20 | 0.05 (0.01) | 0.24 (0.06) | 0.07 (0.01) | 0.30 (0.03) |
| CausalNF | 50 | 1.5 (0.26) | 0.93 (0.13) | 3.09 (0.55) | 0.95 (0.04) |
| Cond-FiP | 50 | 0.08 (0.01) | 0.25 (0.05) | 0.07 (0.00) | 0.48 (0.07) |
| CausalNF | 100 | 1.23 (0.13) | 0.85 (0.08) | 1.67 (0.13) | 0.96 (0.04) |
| Cond-FiP | 100 | 0.07 (0.01) | 0.29 (0.07) | 0.09 (0.01) | 0.57 (0.07) |

*Table 29.* **Results for Sample Generation with CausalNF.** We compare Cond-FiP against CausalNF for the task of generating samples from input noise variables. We find that CausalNF underperforms compared to Cond-FiP by a significant margin.

# H. Evaluating Generalization of Cond-Fip to Larger Sample Size

In the main tables (Table 1, Table 2, and Table 3), we evaluated Cond-FiP's generalization capabilities to larger graphs ($d = 50$, $d = 100$) than those used for training ($d = 20$). In this section, we carry a similar experiment where instead of increasing the total nodes in the graph, we test Cond-FiP on datasets with more samples $n_{\text{test}} = 1000$, while Cond-FiP was only trained for datasets with sample size $n_{\text{train}} = 400$.

The results for the experiments are presented in Table 30, Table 31, and Table 32 for the task of noise prediction, sample generation, and interventional generation respectively. Our findings indicate that Cond-FiP is still able to compete with other baseline in this regime. However, we observe that the performances of Cond-FiP did not improve by increasing the sample size compared to the results obtained for the 400 samples case, meaning that the performance of our models depends exclusively on the setting used at training time. We leave for future works the learning of a larger instance of Cond-FiP trained on larger sample size problems.

| Method | Total Nodes | LIN **IN** | RFF **IN** | LIN **OUT** | RFF **OUT** |
|---|---|---|---|---|---|
| DoWhy | 10 | 0.02 (0.0) | 0.10 (0.01) | 0.21 (0.04) | 0.23 (0.02) |
| DECI | 10 | 0.05 (0.01) | 0.12 (0.01) | 0.21 (0.04) | 0.27 (0.03) |
| FiP | 10 | 0.03 (0.0) | 0.06 (0.0) | 0.21 (0.04) | 0.23 (0.02) |
| Cond-FiP | 10 | 0.05 (0.01) | 0.11 (0.01) | 0.21 (0.04) | 0.25 (0.02) |
| DoWhy | 20 | 0.02 (0.0) | 0.11 (0.02) | 0.16 (0.01) | 0.3 (0.02) |
| DECI | 20 | 0.04 (0.01) | 0.11 (0.02) | 0.16 (0.01) | 0.29 (0.02) |
| FiP | 20 | 0.03 (0.0) | 0.08 (0.02) | 0.16 (0.01) | 0.26 (0.02) |
| Cond-FiP | 20 | 0.06 (0.01) | 0.09 (0.01) | 0.18 (0.01) | 0.26 (0.01) |

*Table 30.* **Results for Noise Prediction with Larger Sample Size** ($n_{\text{test}} = 1000$). We compare Cond-FiP against the baselines for the task of predicting noise variables from the input observations. Each cell reports the mean (standard error) RMSE over the multiple test datasets for each scenario.

| Method | Total Nodes | LIN **IN** | RFF **IN** | LIN **OUT** | RFF **OUT** |
|---|---|---|---|---|---|
| DoWhy | 10 | 0.04 (0.0) | 0.14 (0.02) | 0.29 (0.04) | 0.3 (0.03) |
| DECI | 10 | 0.07 (0.01) | 0.17 (0.02) | 0.29 (0.04) | 0.33 (0.04) |
| FiP | 10 | 0.05 (0.0) | 0.09 (0.01) | 0.29 (0.04) | 0.29 (0.03) |
| Cond-FiP | 10 | 0.05 (0.01) | 0.14 (0.02) | 0.29 (0.04) | 0.29 (0.03) |
| DoWhy | 20 | 0.04 (0.01) | 0.21 (0.05) | 0.28 (0.01) | 0.55 (0.06) |
| DECI | 20 | 0.07 (0.01) | 0.21 (0.04) | 0.29 (0.01) | 0.59 (0.06) |
| FiP | 20 | 0.05 (0.0) | 0.17 (0.04) | 0.28 (0.01) | 0.53 (0.06) |
| Cond-FiP | 20 | 0.05 (0.0) | 0.24 (0.05) | 0.28 (0.01) | 0.53 (0.06) |

*Table 31.* **Results for Sample Generation with Larger Sample Size** ($n_{\text{test}} = 1000$). We compare Cond-FiP against the baselines for the task of generating samples from the input noise variables. Each cell reports the mean (standard error) RMSE over the multiple test datasets for each scenario.

| Method | Total Nodes | LIN **IN** | RFF **IN** | LIN **OUT** | RFF **OUT** |
|---|---|---|---|---|---|
| DoWhy | 10 | 0.04 (0.01) | 0.16 (0.03) | 0.26 (0.03) | 0.27 (0.03) |
| DECI | 10 | 0.09 (0.01) | 0.19 (0.02) | 0.26 (0.03) | 0.31 (0.04) |
| FiP | 10 | 0.05 (0.01) | 0.12 (0.02) | 0.26 (0.03) | 0.27 (0.03) |
| Cond-FiP | 10 | 0.09 (0.02) | 0.19 (0.03) | 0.27 (0.03) | 0.3 (0.03) |
| DoWhy | 20 | 0.04 (0.0) | 0.20 (0.04) | 0.26 (0.01) | 0.53 (0.06) |
| DECI | 20 | 0.08 (0.01) | 0.20 (0.03) | 0.29 (0.02) | 0.54 (0.05) |
| FiP | 20 | 0.06 (0.01) | 0.16 (0.04) | 0.28 (0.02) | 0.48 (0.06) |
| Cond-FiP | 20 | 0.07 (0.01) | 0.27 (0.05) | 0.30 (0.02) | 0.51 (0.06) |

*Table 32.* **Results for Interventional Generation with Larger Sample Size** ($n_{\text{test}} = 1000$). We compare Cond-FiP against the baselines for the task of generating interventional data from the input noise variables. Each cell reports the mean (standard error) RMSE over the multiple test datasets for each scenario.

