# OpenReview forum: "Zero-Shot Learning of Causal Models"
_ICML.cc/2025/Conference — Submitted to ICML 2025_

### Official Review · Reviewer_xGnM · 2025-02-26

**Overall Recommendation:** 4

**Summary:**

Learning the causal generative process from observational data is a challenging problem bottlenecked by the necessity of learning a separate causal model for each dataset. This paper studies a unifying framework to enable zero-shot inference of causal generative processes of arbitrary datasets by training a single model. The authors adapt a recent advancement in causal generative modeling (FiP) to infer generative SCMs conditional on empirical dataset representations in a supervised setup, where the SCM is reformulated as a fixed-point problem. They propose an amortized procedure that takes in a dataset and its causal graph and learns a dataset representation. Then, the authors train a model conditioned on dataset embeddings to learn the functional mechanisms of the generative SCM. This framework enables both observational and interventional sample generation in a zero-shot manner. Empirical results show that the method performs competitively with baseline models for in-distribution and out-of-distribution settings.

## Update After Rebuttal
The authors have done a great job in addressing all of my questions and concerns about this work. Therefore, I am strongly in favor of **acceptance**.

**Claims And Evidence:**

Yes

**Essential References Not Discussed:**

N/A

**Experimental Designs Or Analyses:**

Yes, I checked the results for noise prediction, sample generation, and interventional sampling for all datasets. Furthermore, I checked the out-of-distribution performance.

**Methods And Evaluation Criteria:**

Yes

**Other Comments Or Suggestions:**

N/A

**Other Strengths And Weaknesses:**

## Strengths
- The paper is written well with clear intuitions and explanations as to how it relates to similar work (e.g., FiP).
- Although the assumptions are a bit strong (additive noise model, causal graphs known, noise variables known), the general idea of using an amortized procedure to approximate SCM distributions in a zero-shot manner is quite interesting.
- The empirical results are convincing and show interesting observations, especially the performance of sample generation under distribution shifts and as the causal graphs scale up. It is certainly impressive that Cond-FiP can approximate the SCM distribution of 50 and 100 graph nodes quite well given that it was trained on only datasets with 20-node graph size.

## Weaknesses
- In a synthetic data scenario, assuming access to the noise samples is a feasible assumption, but for real-world datasets, this will not typically hold. Using the noise samples as the supervision for the dataset embedding model may easily become unrealistic. The authors have an evaluation on a real-world benchmark (Sachs) in the appendix where they fit a Gaussian model. However, interventional sample results are not provided.
- The idea to directly infer the SCM distribution under the additive noise model assumption is interesting. However, the feasibility of this assumption may not always hold. It is true that we often parameterize causal models as linear/nonlinear additive noise models, but this can be violated in practice. It seems that this approach would only hold under the strict assumption of additive noise models.
- Knowledge of the causal graph for several datasets can potentially be a strong assumption. In real-world datasets, the causal graph may be unknown and must be discovered. However, for the sake of this work, the synthetic scenarios are good for proof of concept.

**Questions For Authors:**

- Could the authors explain why Cond-FiP performs similar to some baselines in noise prediction and sample generation, especially when the node scale is the same or smaller than used in training? How is the FiP model implemented? In the original paper, it seems that the task is the recovery of the topological ordering. Is the FiP baseline here aware of the causal structure of datasets?
- How does the alternating application of transformer blocks E work? Is this just an alternating optimization method where you optimize for samples when nodes are fixed and optimize for nodes when samples are fixed?
- The main capabilities of the proposed framework are noise prediction and observational/interventional sample generation. However, individual counterfactual sample generation is also important in many applications. Can this framework enable counterfactual inference?

**Relation To Broader Scientific Literature:**

This paper is one of the first to consider generalizing the learning of functional mechanisms of structural causal models from arbitrary datasets and causal graphs and is a significant step toward building causally-aware foundation models.

**Theoretical Claims:**

N/A

---

> ### Author Rebuttal · Authors · 2025-03-31
>
> We thank the reviewer for their positive and insightful feedback! We appreciate your recognition of the soundness of our framework and the diversity of our experiments. We now address the concerns raised by the reviewer below.
>
> >  Access to noise samples
>
> Thank you for raising this point. We agree with the reviewer that most real-world problems do not provide such supervised signals. However, it is important to note that during inference, Cond-FiP does not require access to noise samples. Instead, it only needs the observations (and a predicted or true causal graph) to infer the functional mechanisms. This allows Cond-FiP to be applied to real-world datasets, as demonstrated in our experiment on the Sachs dataset (Appendix C).
>
> An interesting extension of this work would be to explore a semi-supervised setting, where synthetic and real-world data are mixed during training. However, we believe this is outside the scope of the current paper.
>
> > Interventional results on Sachs
>
> Regarding the lack of interventional generation results on the Sachs dataset, the main issue is that Cond-FiP (along with the other baselines considered in this work) only supports hard interventions, whereas the interventional data available for Sachs involves soft interventions (i.e., the exact interventional operations are unknown). As a result, we are unable to provide a comprehensive evaluation of Cond-FiP, or the other baselines, for interventional predictions on Sachs.
>
>
> > Additive Noise (ANM) Assumption
>
> While the ANM assumption may be seen as a limitation, we would like to clarify that our method relies on the ANM assumption only for training the encoder. This is because we need the encoder to predict the noise from the data in order to obtain embeddings, which is simplified under the ANM assumption, as explained in Appendix A.2. However, it is important to emphasize that, while the ANM assumption is required for training the encoder, it is not necessary for training the decoder.
>
> An interesting avenue for future work would be to explore a more general dataset encoding approach, potentially using self-supervised techniques. However, we believe this falls outside the scope of the current work.
>
> > Knowledge of causal graph
>
> We agree with the reviewer that assuming knowledge of the true causal graphs is a strong assumption. However, as outlined in the manuscript (line 406), we can relax this assumption by inferring the causal graphs in a zero-shot manner via state-of-the-art prior works, such as AVICI. In Appendix D, we provide experimental results where we do not assume prior knowledge of the true graphs and infer them via AVICI. These results demonstrate that Cond-FiP can be extended to zero shot infer the full SCMs  using only observational data.
>
>
> > Cond-FiP performance against baselines
>
> First, it is important to note that all baselines use the true causal graphs for a fair comparison with Cond-FiP, and we employ their original implementations. Additionally, the baselines are trained on each test dataset, serving as the gold standard that our zero-shot approach aims to match. In contrast, although Cond-FiP is trained on specific scales, it generalizes well to both smaller and, more importantly, larger instance problems, while maintaining performance comparable to the other baselines. Furthermore, in scarce data regimes, Cond-FiP demonstrates superior generalization (Appendix E).
>
> > FiP implementation
>
> We use the original code provided by the authors from their paper. In their implementation, the authors offer an adaptation of FiP when the causal graph is known. For a fair comparison, we evaluate Cond-FiP against this variant of FiP in our work.
>
> > Alternating application of transformer blocks E
>
> The alternating block transformer is a feedforward neural network that takes as input a tensor of shape $(B,n,d,d_h)$ where $B$ is the batch size, $n$ is the number of samples, $d$ is the number of nodes and $d_h$ is the hidden dimension. In practice, we first permute the second and third dimensions before applying an attention mechanism to perform attention over the sample dimension. Afterward, we permute them back to apply an attention layer over the node dimension.
>
> > Counterfactual inference results
>
> Thank you for raising this point. Cond-FiP is indeed capable of performing counterfactual generation, and we have conducted an experiment to evaluate this. The results can be found via the [anonymous link](https://anonymous.4open.science/r/icml_2025_cond_fip_rebuttal-27D2/Counterfactual_Generation_Exps.pdf). We observe that Cond-FiP performs slightly worse than the baselines in this task, and we believe that improving its performance in counterfactual generation will be a valuable direction for future work.
>
> Thank you once again for your constructive comments! We are open to further discussion and would be happy to address any remaining concerns.

---

> > ### Comment · Reviewer_xGnM · 2025-04-02
> >
> > The authors have given satisfactory clarifications and have provided some new experimental results to evaluate counterfactual inference to address my questions and concerns. Overall, I believe this is a well-written and well-motivated paper that sets forth some interesting ideas for the future of developing robust causally-aware foundation models. Therefore, I keep my rating as Accept.

---

> > > ### Author Response · Authors · 2025-04-08
> > >
> > > Thank you for your support and constructive feedback! We really appreciate your thoughtful comments. Based on the discussion during the rebuttal, we will revise the manuscript accordingly. In particular, we will include the counterfactual inference results in the updated draft, as well as provide more clarifications regarding the other points on interventional results on Sachs dataset, access to noise variables during training, etc.

---

### Official Review · Reviewer_ng7r · 2025-03-09

**Overall Recommendation:** 2

**Summary:**

This paper introduces a method called Cond-FiP for transfer learning of causal mechanisms in causal systems, specifically Structural Causal Models (SCMs). Given the causal variables and their graph, the approach aims to learn a single model capable of inferring the distributions of causal variables without dataset-specific training. Cond-FiP utilizes an encoder to create embeddings of datasets based on observations and causal graphs, and then conditions a fixed-point approach (FiP) decoder on these embeddings to infer the functional mechanisms of SCMs. Experiments are presented to show that Cond-FiP can perform similarly to state-of-the-art methods trained for individual datasets.

## Update after rebuttal

The rebuttal and further discussion with the authors have addressed some of my concerns. However, the specific practical application targeted by this setup remains unclear and the assumptions taken are very strong. Thus, I maintain borderline on this paper.

**Claims And Evidence:**

The paper makes several claims regarding the capabilities of Cond-FiP, and these are generally supported by the evidence presented, primarily through empirical evaluations. The central claim of zero-shot causal mechanism inference is addressed in the experiments by demonstrating that Cond-FiP, trained on a distribution of SCMs, performs sufficiently well on unseen datasets. The claim of achieving performance on par with SoTA methods is supported by the comparative results against baselines like FiP, DECI, and DoWhy across various tasks (noise prediction, sample generation, intervention) and datasets (AVICI, CSuite, Sachs).

The claim of generalizing to out-of-distribution graphs is not well supported. As stated in Section B.1, the noise variables are limited to Gaussian or Laplace distributions. Both distributions have very similar patterns, but no more complex distributions like bi-modal Gaussians, distributions with complex random transformations, etc. have been tested. Thus, the claims are effectively limited to fixed, known noise distributions.

**Essential References Not Discussed:**

Most essential references have been discussed to my knowledge.

**Experimental Designs Or Analyses:**

I have checked the experiments presented in the main paper, most carefully the synthetic data generation. It is generally sound, but limiting in its diversity and out-of-distribution consideration. This limits the claims as mentioned above.

**Methods And Evaluation Criteria:**

Cond-FiP appears to be a sensible approach for the setting considered in the paper. The benchmarks cover standard synthetic and real-world inspired settings.

**Other Comments Or Suggestions:**

Typos:
- Line 155: missing punctuation in "nodes on the current one However, the"

**Other Strengths And Weaknesses:**

The paper misses to make a strong case for the practical application and relevance of the proposed setup. One requires a lot of prior knowledge of the system that one is interested in learning (the causal graph, the general distributions to apply the right model, etc.), where it is further difficult to obtain samples from. For instance, the paper discussed the possible setup of first learning the causal graph with a standard causal discovery approach before applying their method. However, most standard causal discovery approaches require a sufficient number of samples to accurately estimate independence tests or learn the mechanisms themselves.

Further, it is a strong assumption to have access to the noise variables during training. This is generally not possible in the real-world, so the training must be solely performed on simulation data. This requires the simulation to be very accurately matching the distribution and causal relations of the GT model.

The method is restricted to additive noise models. This restricts its applicability to more complex settings. It is unclear how important this assumption is and whether it could be relaxed.

As mentioned above, the term “zero-shot” is oddly used in the context and needs to be justified.

Finally, the prediction of the noise variables introduces problems that the paper does not discussed. In particular, to predict the noise variables from causal variable observations, the map between noise variables and causal variables must be invertible. Otherwise, the noise variables are not unique. Further, no assumption is taken that the noise must follow a certain distribution. As in this setup, any arbitrary invertible transformation can be applied to the noise variables, it is unclear how the model should be able to predict the noise variables.

**Questions For Authors:**

Why are the noise variables predicted? There has been no explicit assumption taken that the map between noise variables and causal variables must be invertible. Further, no assumption is taken that the noise must follow a certain distribution. As in this setup, any arbitrary invertible transformation can be applied to the noise variables, it is unclear how the model should be able to predict the noise variables. How would your method behave on diverse noise distributions with varying complexity?

**Relation To Broader Scientific Literature:**

The definition of "zero-shot" deviates from standard literature and makes the paper's claims confusing. In this paper, zero-shot is defined in line 51 as "zero-shot inference (without updating parameters)", which does not fit in current literature. "Shots" commonly refer to examples that the model sees. In current regimes for LLMs and foundation models, few-shot learning rarely updates the parameters and instead inputs the examples as context. Thus, this paper does not perform "zero-shot" under the current literature. The authors should reconsider whether zero-shot is the best way of terming this setup.

**Theoretical Claims:**

No theoretical claims in the paper.

---

> ### Author Rebuttal · Authors · 2025-03-31
>
> We thank the reviewer for their insightful feedback! We appreciate that they found our claim regarding Cond-FiP’s performance relative to state-of-the-art methods well justified. Additionally, we highlight our experiments in scarce data regimes (Appendix E), where Cond-FiP demonstrates superior generalization compared to baselines.
>
> We now address the reviewer’s concerns below.
>
> > Diverse noise distributions
>
> Thank you for raising this point! Following the reviewer’s recommendation, we experiment with a mixture of gaussian noise for the Large Backdoor and Weak Arrow datasets from CSuite. Specifically, the noise is sampled with equal probability from either $N(0,1)$ or $N(0,2)$. Results accessible via the [anonymous link](https://anonymous.4open.science/r/icml_2025_cond_fip_rebuttal-27D2/GMM_Exps.pdf) show that Cond-FiP is competitive with baselines for sample generation, and slightly worse on other tasks (still competitive with DECI). We emphasize that the baselines are trained from scratch specifically for the mixture of gaussian noise, while Cond-FiP has been pretrained only on gaussian noise.
>
> We also want to clarify the noise distribution choices for the main experiments follow the prior works (Lorch et al. 2022, Scetbon et al. 2024). Finally, our ablation studies in Appendix F.3 also evaluate Cond-FiP’s performance under varying noise distribution complexity. We assess sensitivity to distribution shifts by adjusting noise parameters, controlling the shift magnitude. Results in Tables 25–27 show Cond-FiP’s OOD generalization drops (as expected) as the severity of the shift increases.
>
> > Assumption of known causal graphs
>
> We agree that assuming knowledge of the true causal graphs is a strong assumption. However, standard causal discovery methods are not necessary for inferring causal graphs at inference time. As outlined in the manuscript (line 406), we can relax this assumption by inferring the causal graphs via state-of-the-art amortized causal discovery techniques, such as AVICI. In Appendix D, we provide experimental results where we do not assume prior knowledge of the true graphs and infer them via AVICI. These results demonstrate that Cond-FiP can be extended to infer full SCMs (without updating parameters) using only observational data.
>
> > Additive Noise Model (ANM) assumption
>
> While the ANM assumption may be seen as a limitation, we would like to clarify that our method relies on the ANM assumption only for training the encoder. This is because we need the encoder to predict the noise from the data in order to obtain embeddings, which is simplified under the ANM assumption, as explained in Appendix A.2. However, it is important to emphasize that, while the ANM assumption is required for training the encoder, it is not necessary for training the decoder.
>
> An interesting avenue for future work would be to explore a more general dataset encoding approach, potentially using self-supervised techniques. However, we believe this falls outside the scope of the current work.
>
> > Justification behind predicting noise variables
>
> We agree with the reviewer that the map between the noise variables and causal variables must be invertible. In our setting, we adopt the ANM assumption, which ensures invertibility since the jacobian w.r.t noise is a triangular matrix with nonzero diagonal. Please check Appendix A.1 for more details. Notably, this does not require assuming a specific noise distribution.
>
> > Assumption of noise variables during training
>
> Thank you for raising this point. We agree with the reviewer that most real-world problems do not provide such supervised signals. However, it is important to note that during inference, Cond-FiP does not require access to noise samples. Instead, it only needs the observations (and a predicted or true causal graph) to infer the functional mechanisms. This allows Cond-FiP to be applied to real-world datasets, as demonstrated in our experiment on the Sachs dataset (Appendix C).
>
> An interesting extension of this work would be to explore a semi-supervised setting, where synthetic and real-world data are mixed during training. However, we believe this is outside the scope of the current paper.
>
> > Zero-shot terminology
>
> We agree with the reviewer that the term "zero-shot" in the context of in-context learning and LLM literature typically refers to the number of examples a model sees. However, in our work, we adopt this terminology following the literature on amortized causal learning (Zhang et al. 2023, Nilforoshan et al. 2023, Gupta et al. 2023), where "zero-shot" refers to making predictions without updating the model parameters. We are open to adjusting our notation and adopting the terminology "amortized causal learning" if the reviewer prefers this.
>
> Thank you once again for your constructive comments! We are open to further discussion and would be happy to address any remaining concerns. If you believe your concerns have been addressed, kindly increase your score.

---

> > ### Comment · Reviewer_ng7r · 2025-04-02
> >
> > Thank you for your answers.
> >
> > > Diverse Noise Distributions
> >
> > The chosen mixture of Gaussians is still very similar to a single Gaussian. My question was targeting more complex distributions that significantly differ from the standard Gaussian shape, like a mixture of N(-2,1) and N(2,1)?
> >
> > > Zero-Shot Terminology
> >
> > Thank you, I believe using "amortized causal learning" would be more widely fitting for the goal of this paper.

---

> > > ### Author Response · Authors · 2025-04-04
> > >
> > > Thank you for your response to our rebuttal! We appreciate your feedback and have taken your concerns into account.
> > >
> > > > Diverse Noise Distributions
> > >
> > > Thank you for highlighting this point! Following your recommendation, we have conducted additional experiments with various noise distributions, each modeled as a multi-modal gaussian mixture. Specifically, we considered the following cases:
> > >
> > > - Noise is sampled with equal probability from either $N(-2, 1)$  and $N(2, 1)$.
> > > - Noise is sampled with equal probability from either $N(-2, 2)$  and $N(2, 2)$.
> > > - Noise is sampled with equal probability from either $N(-2, 1)$  and $N(2, 2)$.
> > > - Noise is sampled with equal probability from either $N(-5, 1)$  and $N(5, 1)$.
> > > - Noise is sampled with equal probability from either $N(-5, 2)$  and $N(5, 2)$.
> > > - Noise is sampled with equal probability from either $N(-5, 1)$  and $N(5, 2)$.
> > >
> > > We ran experiments using these $6$ noise distributions on both the Large Backdoor and Weak Arrow datasets from the CSuite benchmarks, leading to a total of $12$ experimental settings. The results, available via this [anonymous link](https://anonymous.4open.science/r/icml_2025_cond_fip_rebuttal-27D2/Multi_Modal_GMM_Experiments.pdf) , demonstrate that Cond-FiP remains competitive with baselines across all tasks. Importantly, while baselines were trained from scratch for each specific gaussian mixture noise distribution, Cond-FiP was pretrained only on gaussian noise and generalizes effectively to these settings.
> > >
> > > > Zero-Shot Terminology
> > >
> > > We appreciate your suggestion and agree that "amortized causal learning" better captures our approach. Since changes to the draft cannot be reflected during the rebuttal phase, we outline below the planned revisions:
> > >
> > > - The title will be updated to "Amortized Learning of Structural Causal Models."
> > > - Phrases such as "we zero-shot infer causal mechanisms" will be revised to "we infer causal mechanisms without any parameter updates."
> > >
> > > Thank you again for your thoughtful and constructive feedback! We will incorporate the gaussian mixture model experiments into the final draft and update our terminology accordingly. If you believe we have satisfactorily addressed your concerns, we would greatly appreciate an increase in your score.

---

### Official Review · Reviewer_dNad · 2025-03-23

**Overall Recommendation:** 3

**Summary:**

This paper addresses the problem of inferring structural causal models (SCMs) from observational data. Unlike previous approaches that train separate models for each observational dataset, this work proposes learning a single model across a distribution of problem instances, enabling zero-shot inference of the underlying SCM. This pipeline comprises of an encoding network that encodes the observed dataset and the underlying graph and a conditional fixed point method that infers SCM conditioned on the observed dataset. Experimental results indicate that the proposed method performs comparably to existing approaches that train individual models for each dataset.

**Claims And Evidence:**

The paper's central claim is that a single model, trained using the proposed pipeline, can perform comparably to training separate models for each dataset. The authors validate this claim through extensive experiments across various problem instances. In each experiment, the reported results indicate that the proposed pipeline performs on par with existing methods (DoWhy, DECI, and FiP) that train distinct models for each dataset.

**Essential References Not Discussed:**

All essential references are discussed to the best of my knowledge.

**Experimental Designs Or Analyses:**

The experiments cover a number of problem instances (linear/non-linear causal relationships, different numbers of nodes, diverse graph structures etc.), a number of relevant benchmarks and relevant metrics for both in and out of distribution evaluation samples. The experiments in the Appendix, particularly the real-world experiment further validates the usefulness of the proposed method. See the questions sections for some concerns regarding the sparse data tables.

**Methods And Evaluation Criteria:**

The proposed method consists of two key components: (1) an encoder that captures information from observations and the underlying causal graph, and (2) a conditional variant of the Fixed-Point Approach (FiP), called Cond-FiP, which infers the SCM conditioned on the encoding from the first step. This approach is benchmarked against existing methods that train separate models for each dataset, namely DoWhy, DECI, and FiP. All methods are evaluated using the MSE metric across three tasks: noise prediction, sample generation, and interventional generation, for both in-distribution and out-of-distribution problem instances.

The proposed methods and/or evaluation criteria makes sense for the problem at hand - this pipeline learns a single model that can perform zero-shot inference of SCMs given observational data and the causal graph for a variety of problem types; the benchmarks cover a number of techniques used in learning distinct models for each dataset; and the various metrics measure how well the SCM was learned based on both the function and noise approximation.

**Other Comments Or Suggestions:**

See questions section.

**Other Strengths And Weaknesses:**

This paper introduces a novel framework that enables amortized learning of causal mechanisms across different instances within the functional class of SCMs. This can help in leveraging shared information between datasets and also has the added benefit of training and storing just a single model for a variety of tasks requiring SCM inference. The paper is very well written, well positioned and the experiments (including the appendix section) is thorough.

The paper demonstrates an average level of originality as it builds on pre-existing ideas like the transformer architecture for SCM inference and FiP. See the questions section for more concerns, particularly about measuring the benefits of this new approach.

**Questions For Authors:**

1. The main benefits of the proposed pipeline lie in learning a single model for inferring various SCMs, as well as the advantages of leveraging shared information. I have two questions regarding the benefits of the framework:
1.1. The paper claims that the proposed method performs well in low-data scenarios. However, Table 11 (Appendix E) suggests that DoWhy often matches or outperforms the proposed approach. Could you provide some insights into why this happens?
1.2. A key advantage of the proposed method is that it learns a single model instead of multiple models. How does this compare in terms of memory requirements, inference time, and computational efficiency?

2. Given that the primary evidence supporting the claims of the paper comes from experiments, are there any plans to open-source the code to facilitate replicability and transparency?

**Relation To Broader Scientific Literature:**

The main contribution of this paper is amortizing the learning of the functional relationships to directly infer the SCMs. Other works either learn a separate model per observational dataset or propose techniques for tasks like amortized causal structure learning, average treatment effect (ATE) estimation etc.

**Theoretical Claims:**

There are no theoretical claims in this paper.

---

> ### Author Rebuttal · Authors · 2025-03-31
>
> We thank the reviewer for their positive and insightful feedback! Thank you for acknowledging the soundness of our framework and diverse experiments. We now address the concerns raised by the reviewer below.
>
> >  The paper claims that the proposed method performs well in low-data scenarios. Table 11 (Appendix E) suggests that DoWhy often matches or outperforms the proposed approach. Could you provide some insights into why this happens?
>
> We agree with the reviewer's observation that Cond-FiP performs comparably to DoWhy in Table 11 for the LIN IN and LIN OUT cases. However, we emphasize that Cond-FiP significantly outperforms DoWhy in the RFF IN and RFF OUT cases, particularly when the total number of nodes is 50 or 100. This strong performance in non-linear settings reinforces our claim that Cond-FiP exhibits superior generalization in scarce data regimes. Note that learning effective solutions with limited data is more challenging for the non-linear functional relationships (RFF IN/OUT) scenario. As a result, DoWhy struggles in these scenarios, while Cond-FiP demonstrates a clear advantage.
>
>
> >  A key advantage of the proposed method is that it learns a single model instead of multiple models. How does this compare in terms of memory requirements, inference time, and computational efficiency?
>
> *Memory Requirements.* We trained Cond-FiP on a single L40 GPU with 48GB of memory (see line 323 in the paper), using an effective batch size of $8$ with gradient accumulation. Below, we outline the detailed memory computation:
>
> - Each batch consists of $n=400$ samples with dimension $d=20$, requiring less than $1$ MiB of data in FP32 precision.
> - Storing the model on the GPU requires under 100 MiB.
> - Our transformer architecture has 4 attention layers, a 256-dimensional embedding space, and a 512-dimensional feedforward network. Using a standard (non-flash) attention implementation, a forward pass consumes approximately 30 GiB of GPU memory.
>
> Compared to the baselines, Cond-FiP has similar memory requirements to DECI and FiP, as all three train neural networks of comparable size. The main exception is DoWhy, which fits simpler models for each node, but this approach does not scale well as the graph size increases.
>
> *Computational Cost.* Like other amortized approaches, Cond-FiP has a higher training cost than the baselines, as it is trained across multiple datasets. While the cost of each forward-pass is comparable to FiP, we trained Cond-FiP over approximately 4M datasets in an amortized manner.  However, Cond-FiP offers a significant advantage at inference time since it requires only a single forward pass to generate predictions, whereas the baselines must be retrained from scratch for each new dataset. Thus, while Cond-FiP incurs a higher one-time training cost, its substantially faster at inference.
>
> > Given that the primary evidence supporting the claims of the paper comes from experiments, are there any plans to open-source the code to facilitate replicability and transparency?
>
> Thanks for this point! We plan to open-source the code along with comprehensive documentation to facilitate reproducibility of our experiments. For the rebuttal phase, we have prepared an anonymized version of the codebase, which can be accessed via this [link](https://anonymous.4open.science/r/icml_2025_cond_fip_rebuttal-27D2/).
>
> Please note that while the codebase is not directly executable, it provides full access to the implementation of all components of our framework:
>
> -  `cond_fip/models` contains the implementation of the transformer-based encoder and the Cond-FIP architecture.
> - `cond_fip/tasks` includes the training and inference methods associated with our framework.
>
> > The paper demonstrates an average level of originality as it builds on pre-existing ideas like the transformer architecture for SCM inference and FiP.
>
> We agree with the reviewer that our framework builds upon prior works, specifically AVICI (Lorch et al. 2022) and FiP (Scetbon et al. 2024). However, we want to clarify that our main contribution relies on integrating these two frameworks to enable zero-shot inference of generative SCMs, a problem not previously addressed. To achieve this, we made substantial modifications to the FiP architecture, as it needed to be conditioned on datasets. Additionally, to facilitate this dataset conditioning we propose to learn dataset embeddings via the noise prediction task.
>
> While most existing studies on amortized causal learning focus on treatment effect estimation or causal discovery (as discussed in Section 2), our work tackles the novel task of amortized learning to infer the causal mechanisms of SCMs.
>
> Thanks again for your constructive comments! We are open to further discussion and would be happy to address any remaining concerns.

---

### Decision · Program_Chairs · 2025-05-01

**Decision:**

Reject

**Comment:**

This paper aims to amortize the inference of structural causal models (SCMs) from observational data; it learns a single model across a distribution of problem instances to enable zero shot prediction of the underlying SCM. The method encodes the observed data and uses a conditional fixed point method to infer the SCM conditioned on the observed dataset. The results indicate that the method performs comparably relative to existing approaches that train individual models for each dataset. The idea of amortizing structure estimation is not new and in the rebuttal the authors note that the paper is a combination of AVICI (Lorch et al. 2022, which studies amortizing the problem of structure estimation) and FiP (Scetbon et al. 2024, a neural network architecture that represents SCMs using fixed point operations). While I do think the authors do a good job of responding to reviewer concerns, at the end of the rebuttal period the paper remained borderline in large part because there was no well motivated practical use-case for the idea identified. In addition I think some ways to strengthen the paper include an empirical study of how identifiability of the underlying graph might influence the capabilities --e.g. does the size of the markov equivalence class change the hardness of amortization as well as how uncertainty in the prediction correlates with the size of the equivalence class given a finite sample of data.